# Near-Optimal Rates for $O(1)$-Smooth DP-SCO with a Single Epoch and Large Batches

**Christopher A. Choquette-Choo**          CCHOQUETTE@GOOGLE.COM
*Mountain View, CA, USA*

**Arun Ganesh**          ARUNGANESH@GOOGLE.COM
*Seattle, WA, USA*

**Abhradeep Thakurta**          ATHAKURTA@GOOGLE.COM
*Mountain View, CA, USA*

**Editors:** Gautam Kamath and Po-Ling Loh

## Abstract

In this paper we revisit the DP stochastic convex optimization (SCO) problem. For convex smooth losses, it is well-known that the canonical DP-SGD (stochastic gradient descent) achieves the optimal rate of $O\left(\frac{LR}{\sqrt{n}} + \frac{LR\sqrt{p\log(1/\delta)}}{\varepsilon n}\right)$ under $(\varepsilon, \delta)$-DP (Bassily et al., 2014), and also well-known that variants of DP-SGD can achieve the optimal rate in a single epoch (Feldman et al., 2020). However, the batch gradient complexity (i.e., number of adaptive optimization steps), which is important in applications like federated learning, is less well-understood. In particular, all prior work on DP-SCO requires $\Omega(n)$ batch gradient steps, multiple epochs, or convexity for privacy.

We propose an algorithm, `Accelerated-DP-SRGD` (stochastic recursive gradient descent), which bypasses the limitations of past work: it achieves the optimal rate for DP-SCO (up to polylog factors), in a single epoch using $\sqrt{n}$ batch gradient steps with batch size $\sqrt{n}$, and can be made private for arbitrary (non-convex) losses via clipping. If the global minimizer is in the constraint set, we can further improve this to $n^{1/4}$ batch gradient steps with batch size $n^{3/4}$. To achieve this, our algorithm combines three key ingredients, a variant of stochastic recursive gradients (SRG), accelerated gradient descent, and correlated noise generation from DP continual counting.

## 1. Introduction

Differentially private stochastic convex optimization (DP-SCO) and its close counterpart, DP empirical risk minimization (DP-ERM) are two of the most well studied problems in private machine learning (Bassily et al., 2014, 2019; Feldman et al., 2020; Bassily et al., 2020; Asi et al., 2021a; Bassily et al., 2021; Zhang et al., 2022b; Asi et al., 2021b; Kulkarni et al., 2021; Gopi et al., 2022; Chaudhuri et al., 2011; Kifer et al., 2012). Optimal rates are known for nearly all classes of convex losses, e.g., Lipschitz (and smooth) (Bassily et al., 2019; Feldman et al., 2020), and Lipschitz with strong convexity (Gopi et al., 2022). There also have been extensive work in the recent past (Kulkarni et al., 2021; Feldman et al., 2020; Gopi et al., 2022; Gao et al., 2024; Zhang et al., 2022a) optimizing for both the number of gradient oracle calls (Bubeck, 2015) and rounds of adaptive interactions (Smith et al., 2017; Woodworth et al., 2018) while achieving optimal DP-SCO error.

In this work, we revisit the problem of reducing both the gradient complexity, and the number of adaptive interactions (a.k.a. *batch gradient steps*) for smooth[1] and convex losses. While this problem is well-understood in the non-private setting Carmon et al. (2023); Jambulapati et al. (2024),

---

[1] Unless mentioned otherwise, we refer to $O(L/\|\mathcal{C}\|_2)$ $\ell_2$-smoothness. For brevity, we will sometimes refer to this as $O(1)$-smoothness as this gives smoothness parameter $M = O(1)$ if $L, \|\mathcal{C}\|_2 = \Theta(1)$, and $M = O(L/\|\mathcal{C}\|_2)$ is within a constant of the lower bound $M \geq L/\|\mathcal{C}\|_2$ that holds if $\mathcal{C}$ contains the global minimizer.

little work has been done on it in the private setting. We show that it is possible to obtain optimal DP-SCO error (upto polylog$(n, 1/\delta)$ factors), in a single epoch algorithm with sublinear batch gradient steps.

**Problem definition:** There is a distribution of examples $\mathcal{D}$ and a loss $f(\mathbf{x}, d)$ where $\mathbf{x} \in \mathcal{C} \subseteq \mathbb{R}^p$ is a model in the constraint set $\mathcal{C}$, $\|\mathcal{C}\|_2 \leq R$, and $d$ is an example. We assume for any $d \in supp(\mathcal{D})$, $f$ is convex, $L$-Lipschitz (i.e. $\|\nabla f(\mathbf{x}; d)\|_2 \leq L$ for all $\mathbf{x}, d$, and $M$-smooth (i.e. $\|\nabla f(\mathbf{x}; d) - \nabla f(\mathbf{y}; d)\|_2 \leq M \|\mathbf{x} - \mathbf{y}\|_2$ for all $\mathbf{x}, \mathbf{y}, d$) over $\mathcal{C}$. We assume for some $c_M = O(1)$, we have $M \leq c_M L / \|\mathcal{C}\|_2$. We receive a sequence of $n$ i.i.d. examples $D = \{d_1, d_2, \ldots, d_n\} \sim \mathcal{D}$. Using these examples, our goal is to choose a model in $\mathcal{C}$ that minimizes the population risk function $F(\mathbf{x}) = \mathbb{E}_{d \sim \mathcal{D}} [f(\mathbf{x}, d)]$. We will operate in the single-pass streaming setting: we must process examples in the order they appear, and each example appears only once in this order. Finally, the model we output should satisfy $(\varepsilon, \delta)$-DP w.r.t. the examples. For simplicity, we use the zero-out notion of DP (Ponomareva et al., 2023): if $P$ is the distribution of models our algorithm outputs given $\{d_1, d_2, \ldots, d_n\}$ and $Q$ is the distribution given the same sequence, except one example $d_i$ instead has $\nabla f(\mathbf{x}, d_i) = 0$ for all $\mathbf{x}$, then we require $\max_S [P(S) - e^\varepsilon Q(S)], \max_S [Q(S) - e^\varepsilon P(S)] \leq \delta$.

**Our result:** We show the following:

1. **Realizable setting**: Assuming the realizable setting (that is, the unconstrained population risk minimizer is in the interior of the constraint set, i.e. $\mathbf{x}^\dagger = \arg\min_{\mathbf{x} \in \mathbb{R}^p} F(\mathbf{x})$, and $\mathbf{x}^\dagger \in \mathcal{C})^2$, *there exists a gradient based DP algorithm that makes a single pass over the data set (a.k.a. single epoch) and achieves optimal DP-SCO error (within polylog factors) for smooth and convex losses, with $T = \widetilde{\Theta}(n^{1/4})$ batch gradient steps[3], batch size $B = n/T$, and using exactly two gradients per training example.*

2. **Non-realizable setting**: We also show that even in the non-realizable setting, *our algorithm obtains the optimal DP-SCO bounds with $T = \sqrt{n}$ batch gradient steps, batch size $B = \sqrt{n}$ and in a single epoch.*

Our work uniformly improves over all prior (Feldman et al., 2020; Zhang et al., 2022b,a) and contemporary (Gao et al., 2024) works *in the realizable setting*. Under smoothness assumption, prior SoTA (in terms of batch gradient complexity) required gradient complexity of $\widetilde{\Theta}(n)$, and with batch gradient complexity of $T = \widetilde{\Theta}(\sqrt{n})$. Under a single-epoch constraint, to the best of our knowledge no prior work achieves $T = o(n)$. In comparison, our work improves the gradient complexity to $2n$, and batch gradient steps to $T = \widetilde{\Theta}(n^{1/4})$. A contemporary work (Gao et al., 2024) matches our batch gradient steps ($T = \widetilde{\Theta}(n^{1/4})$). However, they have worse gradient complexity of $\widetilde{\Theta}(n^{9/8})$.

Our guarantees, even in the non-realizable setting, *improves the batch gradient complexity over all prior work* (at least by polylog$(n)$), and is incomparable to the contemporary work of Gao et al. (2024). (See Table 1.)

Works of (Feldman et al., 2020; Carmon et al., 2023; Gao et al., 2024) also study DP-SCO with higher smoothness conditions of e.g. $M = O(\sqrt{n})$, which by Nesterov smoothing (Nesterov,

---

[2]In the unconstrained optimization setting, a standard assumption is a known bound $R$ on the norm of $\mathbf{x}^\dagger$ and we can set $\mathcal{C} = B(\mathbf{0}, R)$ to satisfy our assumption. Hence, our assumption can be viewed as implying an unconstrained optimization of the population risk $F(\mathbf{x})$. A similar assumption was made in (Xiao, 2009, Theorem 8) while designing accelerated version of dual averaging methods.

[3]Here, $n$ is the number of training examples.

2005) corresponds to the non-smooth setting. However, these works require $n^{\Omega(1)}$ passes over the dataset; to the best of our knowledge there is no result in the literature giving optimal rates for DP-SCO in $n^{o(1)}$ epochs in the non-smooth setting (for all settings of the dimension $p$; (Carmon et al., 2023) uses $O(1)$ epochs if $p \leq n^{1/2}\varepsilon^{3/2}$). Since we are interested in the *single-epoch setting*, we hence only focus on $O(1)$-smoothness, and higher smoothness is beyond scope. As an aside, in the $M = O(\sqrt{n})$ regime, (Carmon et al., 2023) achieves worst-case gradient complexity of $\widetilde{\Theta}(n^{4/3})$, as opposed to $\widetilde{\Theta}(n^{11/8})$ by (Gao et al., 2024).

**Gradient complexity and adaptive interaction complexity (a.k.a. batch gradient steps):** We measure the computational complexity of our algorithms under standard complexity measures in the convex optimization literature: a) *Gradient complexity* (Bubeck, 2015): It measures the total number of gradient evaluations of the loss function $f(\mathbf{x}, \cdot)$, and b) *Batch gradient steps (a.k.a. adaptive interaction complexity)* [4] (Smith et al., 2017; Woodworth et al., 2018): It corresponds to the number of adaptive gradient computations ($T$) the algorithm has to perform, i.e., the current computation depends on the output of the gradient computations in the previous state. This is equivalent to the model introduced by Nemirovski (1994) where one has access to many parallel gradient oracles. Understanding the complexity of an algorithm in terms of number of batch gradient steps have significant attention in the recent past (Smith et al., 2017; Woodworth et al., 2018; Diakonikolas and Guzmán, 2020; Carmon et al., 2023; Jambulapati et al., 2024). It is especially important in the context of distributed/federated optimization, where adaptive interactions are much more expensive compared to parallel gradient evaluations at a fixed model state. For example, in cross-device federated learning (Kairouz et al., 2021b), the number of rounds adaptive server/device communication dominate the overall optimization complexity. Furthermore, our algorithm has additional desirable properties, which are crucial in any practical deployment:

(a) *Single pass/single epoch*: A gradient based algorithm $\mathcal{A}(D) \to \{s_1, \ldots, s_T\}$ is called single epoch, if it computes the gradients on any data point $d \in D$ at exactly one state $s_t$. Without additional restrictions, a state is allowed to encode arbitrary amount of history, i.e., $s_t \supseteq \{s_1, \ldots, s_{t-1}\}$. Single epoch [5] algorithms are crucial in learning over data streams (where you can process any data point at any single model state), or in cross-device federated learning (FL) where a device holding a specific data point appears only once during the training (Balle et al., 2020).

It is worth emphasizing that there is a *crucial difference between an algorithm using a single epoch and an algorithm that performs even two epochs over the data; one may be implementation wise infeasible* because of the reasons stated above. So, in Table 1 even if the gradient complexity matches asymptotically between two algorithms, still one may be infeasible to implement due to the fact of not being single epoch.

(b) *Assuming convexity for privacy*: In DP-SCO literature (see Table 1), prior works have assumed that the loss function $f(\mathbf{x}, d)$ is convex to prove both the privacy and utility guarantee of their algorithms. In these works, convexity is used to enable privacy amplification by iteration Feldman et al. (2018) as gradient steps are contractive, or allows one to use techniques like output perturbation Lowy and Razaviyayn (2024) which assume the loss function's minimizer is stable (perhaps after adding a regularizer). Assuming convexity for differential

---

[4]Henceforth, we will refer to it as batch gradient steps for consistency.

[5]Notice that the definition does not include any restriction on the memory of the algorithm. While stating our results in the single epoch setting, we additionally incorporate the batch size, and the total number gradients evaluated per data point.

| Algorithms | Gradient complexity | Batch gradient steps | Single epoch | Convexity for privacy |
|---|---|---|---|---|
| Ours (assuming $\mathbf{x}^\dagger \in \mathcal{C}$) | $2n$ | $\widetilde{\Theta}(n^{1/4})$ | Yes | No |
| Ours (no assumption) | $2n$ | $\sqrt{n}$ | Yes | No |
| (Feldman et al., 2020) | $n$ | $n/\log^{O(1)}(n)$ | Yes | Yes |
| (Zhang et al., 2022b) | $2n$ | $n$ | Yes | No |
| (Zhang et al., 2022a) | $\widetilde{\Theta}(n)$ | $\widetilde{\Theta}(\sqrt{n})$ | No | Yes |
| (Gao et al., 2024) (Contemporary) | $\widetilde{\Theta}(n^{9/8})$ | $\widetilde{\Theta}(n^{1/4})$ | No | No |

Table 1: Comparison of gradient complexity and batch gradient steps. Here, $\mathbf{x}^\dagger = \arg\min_{\mathbf{x} \in \mathbb{R}^p} F(\mathbf{x})$ is the unconstrained population risk minimizer, and the Lipschitz constant ($L$) and the smoothness required for all results is $O(1)$. $\widetilde{\Theta}(\cdot)$ hides polylog($n$) factors.

privacy guarantees extremely restricts the applicability of the algorithms in practical scenarios where provable convexity property may not hold (e.g., training non-convex models with DP-SGD (Abadi et al., 2016), or simply clipping the gradients to a fixed $\ell_2$-norm (Song et al., 2020).)

Surprisingly, our algorithms, even being single epoch and not assuming convexity for privacy, uniformly beat the prior SoTA in terms of gradient complexity and batch gradient steps.

**Our algorithm [`Accelerated-DP-SRGD` (DP stochastic recursive gradient descent)]:** In terms of design, our algorithm is based on a careful adaptation of *Nesterov's accelerated stochastic gradient descent* (Nesterov, 1983; Bansal and Gupta, 2019), combined with the ideas of *stochastic recursive gradients* (SRG) (Nguyen et al., 2017; Zhang et al., 2022b), and *private continual counting mechanisms* (Dwork et al., 2010; Chan et al., 2011; Denisov et al., 2022; Matoušek et al., 2020; Fichtenberger et al., 2022; Henzinger et al., 2023, 2024; Andersson and Pagh, 2023). In the hindsight, `Accelerated-DP-SRGD` is arguably more natural than prior/contemporary adaptations of accelerated methods (Gao et al., 2024; Zhang et al., 2022a) towards reducing the computational complexity of obtaining optimal DP-SCO guarantees.

Recall, Nesterov's acceleration can be expressed as the coupling of two gradient descent based algorithms with different learning rates. More formally, the following are the update rules[6]: a) $\mathbf{z}_{t+1} \leftarrow \Pi_{\mathcal{C}}\left(\mathbf{z}_t - \frac{\eta_t}{\beta}\nabla_t\right)$, b) $\mathbf{y}_{t+1} \leftarrow \Pi_{\mathcal{C}}\left(\mathbf{x}_t - \frac{1}{\beta}\nabla_t\right)$, and c) $\mathbf{x}_{t+1} \leftarrow (1 - \tau_{t+1})\mathbf{y}_{t+1} + \tau_{t+1}\mathbf{z}_{t+1}$. In our implementation of the Nesterov's method, we will use a DP method to approximate the gradient computed at $\mathbf{x}_t$ computed on a minibatch $\mathcal{B}_t$ of size $B$: $\nabla_t(\mathbf{x}_t) = \frac{1}{B}\sum_{i \in \mathcal{B}_t} \nabla f(\mathbf{x}_t; d_i)$. We use the idea of stochastic recursive gradients (SRGs) to first approximate $\nabla_t$ as $\eta_t \nabla_t \approx \Delta_t + \Delta_{t-1} + \ldots$, where each $\Delta_t = \eta_t \nabla_t(\mathbf{x}_t) - \eta_{t-1}\nabla_t(\mathbf{x}_{t-1})$. SRGs have been used in non-private and private optimization as a variance-reduction method before by Nguyen et al. (2017); Asi et al. (2021a); Bassily et al. (2021); Zhang et al. (2022b); Zhang et al. (2022b)'s usage of SRGs is most similar to ours. We set $\eta_t = t$, $\beta \approx M$ if we assume $\mathbf{x}^\dagger \in \mathcal{C}$ and $\beta \approx M \cdot \max\left\{\frac{n^{3/2}}{B^2}, \frac{n}{B}\right\}$ otherwise, and $\tau_t = \frac{\eta_t}{\sum_{\tau \le t} \eta_\tau}$. Following are the few crucial differences from Zhang et al. (2022b): a) We use a truly accelerated

---

[6]Here, $\Pi_{\mathcal{C}}(\cdot)$ is the $\ell_2$-projection onto to the constraint set, $\{\eta_t\}$ and $\beta$ are scaling parameters, and $\{\tau_t\}$ are the coupling parameters.

method to achieve the improved batch gradient steps, as opposed to tail-averaged online convex optimization methods (which does not include the term $\mathbf{y}_t$), and b) because of acceleration, we cannot tolerate higher values of $\eta_t$. In particular the results of Zhang et al. (2022b) hold with $\eta_t = t^k$ (for any $k \geq 1$). To bound the sensitivity of $\Delta_t$ for privacy, we need to either (i) assume $\mathbf{x}^\dagger \in \mathcal{C}$, which allows us to show the gradient steps are decreasing over time and hence the gradient difference between $\mathbf{x}_t$ and $\mathbf{x}_{t-1}$ is small by smoothness, or (ii) slow down the learning by choosing $\beta \approx M\sqrt{n}$ (corresponding to the iterations $T = n/B \approx \sqrt{n}$), as opposed to $\beta = M$ in the non-private case.

**Our analysis:** While we argue above that our algorithm is a natural extension of Nesterov's accelerated SGD, the privacy and the utility analysis of the algorithm is novel, and far from immediate. We give a brief sketch of our analysis below.

At a high-level our analysis requires bounding the $\ell_2$-sensitivity of the gradient difference $\Delta_t$ to $O(\frac{1}{B \cdot t})$, and then use the binary tree mechanism (Dwork et al., 2010) to compute DP variants of each $\eta_t \nabla_t$ with low error. We finally get an SCO error of the form: $\frac{LR}{\sqrt{n}} + \frac{LR\sqrt{p}}{\varepsilon n} + \frac{LRB}{n} + \frac{LRB\sqrt{p}}{\varepsilon n^2}$, with $B$ being the minibatch size. The first two terms match the optimal DP-SCO error. Setting $B \leq \sqrt{n}$ gives that the third and fourth term also are at most the optimal DP-SCO error. The main challenge is bounding the sensitivity of term $\Delta_t$ (stated above). Bounding it naively to $O\left(\frac{tM}{B\beta} + L\right)$ (using the analysis of Zhang et al. (2022b)), may result in a sub-optimal choice of the batch size for the final DP-SCO guarantee. Instead we couple the privacy analysis and the utility analysis together.

Recall, $\mathbf{x}^\dagger = \arg\min_{\mathbf{x} \in \mathbb{R}^p} F(\mathbf{x})$ is the unconstrained minimizer of the population risk $F(\mathbf{x})$, and we assume the realizable setting i.e., $\mathbf{x}^\dagger \in \mathcal{C}$. We observe that $\|\nabla F(\mathbf{x})\|_2^2 \leq 2M(F(\mathbf{x}) - F(\mathbf{x}^\dagger))$ for any $\mathbf{x} \in \mathcal{C}$. This allows us to get a tighter control over the norm of the difference in mini-batch gradients: $\frac{1}{B} \cdot \left\|\sum_{d \in \mathcal{B}} \nabla f(\mathbf{x}_t; d) - \sum_{d \in \mathcal{B}} \nabla f(\mathbf{x}_{t-1}; d)\right\|_2$, rather than bounding it crudely by $\frac{M}{B\beta}$. As the algorithm proceeds and the excess population risk $F(\mathbf{x}) - F(\mathbf{x}^*)$ goes down, $\left\|\frac{1}{B}\sum_{d \in \mathcal{B}} \nabla f(\mathbf{x}_t; d)\right\|_2$ becomes smaller. This in-turn can be used to obtain a stronger bound of $O\left(\frac{M}{B \cdot \beta t}\right)$, on the norm of the difference in mini-batch gradients. We use this to reduce the sensitivity of $\Delta_t$ to $O\left(\frac{M}{B\beta} + L\right)$. This is formalized via Lemma 7.

If we are in the non-realizable setting, i.e. we do not assume that $\mathbf{x}^\dagger \in \mathcal{C}$, then we proceed with the naive bound on the gradient difference, and compensate by increasing $\beta$, which in turn requires more iterations for the non-private algorithm to converge and hence a smaller batch size of $\Omega(\sqrt{n})$ (as opposed to $B = \widetilde{\Omega}(n^{3/4})$). This still improves on prior work at least by logarithmic factors. More importantly, we still keep the algorithm single epoch and do not require convexity, which is not possible with the comparable prior work of Zhang et al. (2022a).

**Comparison to relevant prior work:** While DP-SCO has been studied extensively in the literature, we first focus on the works that are most relevant for this paper (Feldman et al., 2020; Zhang et al., 2022b; Gao et al., 2024; Zhang et al., 2022a). As summarized in Table 1, our bounds are uniformly better than prior and contemporary works in the realizable setting (i.e., $\mathbf{x}^\dagger \in \mathcal{C}$). Outside this setting, our algorithm is still uniformly better than prior works, and incomparable to the contemporary work of Gao et al. (2024). Unlike many of the prior works, our algorithm is single epoch, and does not require convexity assumption to ensure differential privacy.

In terms of techniqiue, our algorithm heavily relies on the idea of coupling DP-continual observation (Chan et al., 2011; Dwork et al., 2010; Choquette-Choo et al., 2022; Henzinger et al., 2023)

with stochastic recursive gradients (Nguyen et al., 2017) from (Zhang et al., 2022b). However, there are a few crucial differences which allows us to handle much larger batch sizes while still being a single epoch algorithm. Zhang et al. (2022b) and their analysis relies on the access to a low-regret online convex optimization algorithm (OCO) (Hazan, 2019). We know that the regret guarantees of OCO methods cannot be improved beyond $\Omega(1/\sqrt{T})$ even under smoothness (Hazan, 2019). Hence, it is not obvious how the online-to-batch framework (Zhang et al., 2022b) can be used to achieve the number of batch gradient steps of our algorithm. One might think of replacing the OCO algorithm in Zhang et al. (2022b) with a SGD method tailored to smooth losses (Bubeck, 2015, Theorem 6.3) to get around the reduced convergence rate of $\Omega(1/\sqrt{T})$. In Section 4.3 we show that unfortunately, because of the variance introduced by the SRGs, in the absence of an accelerated SGD method, the optimal batch size is still $O(1)$. The fact that we use Nesterov's acceleration allows us to handle much larger batch sizes.

**Other related work:** We discuss some of the prior work (Carmon et al., 2023; Bassily et al., 2021; Asi et al., 2021a; Ganesh et al., 2023). Although the settings/problems in these papers are different than the one we study, all are important for the broader question of adaptive oracle complexity for optimal DP-SCO.

Carmon et al. (2023) provides the best known gradient oracle complexity for optimal DP-SCO, for non-smooth convex losses under $\ell_2$-geometry. While their algorithm is not single-epoch, their overall gradient complexity (i.e., the number of gradient oracle calls) is $\widetilde{O}\left(\min\left\{n, \frac{n^2}{p}\right\} + \min\left\{\frac{(np)^{2/3}}{\varepsilon}, n^{4/3}\varepsilon^{1/3}\right\}\right)$. This result is incomparable to our result, since we rely on smoothness in our analysis, but our overall batch gradient oracle complexity is an improved $\widetilde{O}\left(\sqrt{n}\right)$. Under constant $\varepsilon$, under all regimes of the dimensionality where the DP-SCO guarantee is meaningful i.e., $p \leq n^2$, our gradient oracle complexity is superior. We leave the possibility of removing the smoothness assumption in our paper while obtaining the same (or better) gradient oracle complexity as an open problem. It is also worth mentioning that the results in Carmon et al. (2023) heavily rely on privacy amplification by sampling (Kasiviswanathan et al., 2008; Bassily et al., 2014), and convexity of the loss function for privacy. Our work does not rely on either. At the heart of Carmon et al. (2023), is a new variance bounded stochastic gradient estimator ReSQue. It is an important open question if this estimator can be used to improve the variance properties of our stochastic recursive gradients (SRGs).

(Bassily et al., 2021; Asi et al., 2021a; Ganesh et al., 2023) have looked at various extensions to the $\ell_2$-DP-SCO problem. For example, DP-SCO guarantees under the loss functions being $\ell_1$-Lipschitz, or SCO properties of second-order-stationary points. These results, while being important for the DP-SCO literature, are not comparable to our work.

**Future directions:** While our work significantly improves on the SoTA in terms of gradient complexity and batch gradient steps in DP-SCO for smooth convex losses, it raises a lot of interesting open questions:

(a) Can we obtain the same tighter batch gradient steps guarantee in Table 1, without assuming the unconstrained minimizer to be in the set $\mathcal{C}$? Informally, we only use this assumption to show the gradient norm is decreasing, i.e. the distance between $\mathbf{x}_t - \mathbf{x}_{t-1}$ is decreasing in $t$, and so the gradient differences $\Delta_t$ are bounded by smoothness. However, if gradients remain large during training, we expect training to reach the boundary of $\mathcal{C}$ quickly, in which case $\mathbf{x}_t - \mathbf{x}_{t-1}$ should also be small (because the gradient step is mostly reversed by the projection into $\mathcal{C}$). Hence we believe this assumption is not required for `Accelerated-DP-SRGD` to achieve $T = n^{1/4}$, but we do not know how to formally show this.

(b) In our paper we only focused on $O(1)$-smooth convex losses. To the best of our knowledge, there is no single-epoch algorithm for the non-smooth setting. The best result we are aware of in this setting is that of Carmon et al. (2023), which uses $O(n)$ gradient oracle calls when e.g. $p, \varepsilon$ are constant, but can require up to $O(n^{4/3})$ gradient oracle calls for higher dimensions. It remains an open question what the smallest dimension-independent gradient complexity and batch gradient complexity required for optimal DP-SCO bounds in the non-smooth setting is. Our approach seems to require smoothness as the main primitive, stochastic recursive gradients, cares about differences of gradients at different points. So it is unlikely our techniques can be used to achieve similar improvements in the non-smooth setting.

(c) Finally, while we believe the batch gradient steps bound of $n^{1/4}$ is tight for DP-SCO, we do not know how to prove a lower bound. Standard lower bounding techniques from the non-private literature (Woodworth et al., 2018) for batch gradient steps break down under DP setting as they operate in the regime where the dimensionality $p = \Omega(n^2)$. In such high-dimensional regime, DP-SCO guarantees are vacuous. A well known result of Nesterov (2004) states that that no first-order method (i.e., algorithm that is only allowed to make gradient queries to points in the span of all gradients received so far) can achieve an empirical loss bound of $o(1/T^2)$ in $T$ gradient queries for convex smooth losses, which even ignoring privacy suggests $T = n^{1/4}$ iterations are needed for any algorithm's empirical loss to match the optimal stochastic error $1/\sqrt{n}$. It would be surprising if a DP algorithm could achieve optimal rates in $o(n^{1/4})$ iterations, as in the limit when $\varepsilon \to \infty$ it would beat this non-private lower bound. However, it is not obvious how to extend this lower bound to DP algorithms, as the noise addition violates the constraint that only points in the span of gradients are queried.

## 2. Background on (Private) Learning

### 2.1. `DP-SGD` : DP stochastic gradient descent

`DP-SGD` (Song et al., 2013; Bassily et al., 2014; Abadi et al., 2016) is a standard procedure for optimizing a loss function $\mathcal{L}(\mathbf{x}; D) = \frac{1}{n} \sum_{i \in [n]} \nabla \ell(\mathbf{x}; d_i)$ under DP. The algorithm is as follows: $\mathbf{x}_{t+1} \leftarrow \Pi_{\mathcal{C}} (\mathbf{x}_t - \eta \cdot (\nabla_t + \mathbf{b}_t))$, where $\nabla_t = \frac{1}{|\mathcal{B}|} \sum_{d \in \mathcal{B}} \texttt{clip} (\nabla \ell(\mathbf{x}_t; d), L)$ is the gradient computed on a minibatch of samples in $\mathcal{B} \subseteq D$ at $\mathbf{x}_t$, $\mathbf{b}_t$ is a well-calibrated (independent) spherical Gaussian noise to ensure DP, and $\texttt{clip} (\mathbf{v}, L) = \mathbf{v} \cdot \min \left\{ \frac{L}{\|\mathbf{v}\|_2}, 1 \right\}$. One can show that under appropriate choice of parameters `DP-SGD` achieves the optimal DP-SCO error (Bassily et al., 2019, 2020).

### 2.2. `DP-FTRL` : DP follow the regularized leader

`DP-FTRL` (Smith and Thakurta, 2013; Kairouz et al., 2021a; Denisov et al., 2022) is another standard approach for optimization, where in its simplest form, the state update is exactly that of `DP-SGD` when $\mathcal{C} = \mathbb{R}^p$, except the noise in two different rounds $\mathbf{b}_t, \mathbf{b}_{t'}$ are not independent. Instead (in the one-dimensional setting) $(\mathbf{b}_1, \mathbf{b}_2, \dots \mathbf{b}_T) \sim \mathcal{N}(0, (\mathbf{C}^\top \mathbf{C})^{-1})$, where $\mathbf{C}$ is chosen to minimize the error on prefix sums of $\mathbf{x}_t$. A canonical and asymptotically optimal implementation of `DP-FTRL` is the binary tree mechanism of (Dwork et al., 2010; Chan et al., 2011). While `DP-FTRL` has strong empirical properties comparable to `DP-SGD`, it is unknown whether it can

achieve optimal DP-SCO error unless under additional restrictions (e.g., realizability (Asi et al., 2023)). A line of work (e.g. (Choquette-Choo et al., 2022; Henzinger et al., 2023, 2024)) has looked at optimizing the choice of **C**. These works offer constant-factor improvements over the binary tree mechanism. While these improvements are meaningful practically, we aim for asymptotic guarantees in this paper, and so we will restrict our choice of correlated noise to the binary tree mechanism for simplicity.

## 3. Our Algorithm

In the following we provide our main algorithm (Algorithm 1, `Accelerated-DP-SRGD`) treating the addition of noise for DP as a black box, and giving a utility analysis as a function of the noise. In Section 5.2 we will specify the noise mechanism and its privacy guarantees, and show the resulting utility analysis gives the desired optimal rates.

Our algorithm is a DP variant of Nesterov accelerated SGD (Nesterov, 1983; Bansal and Gupta, 2019) using *stochastic recursive gradients (SRGs)*: similarly to (Nguyen et al., 2017; Zhang et al., 2022b) we maintain $\nabla_t$, an unbiased estimate of $\nabla F(\mathbf{x}_t)$, and use $\nabla_t$ and batch $t+1$ to obtain $\nabla_{t+1}$. Our algorithm incorporates noise $\mathbf{b}_t$ for DP. We first give a utility analysis for a fixed value of the noise $\mathbf{b}_t$, and then in Section 5.2 we will specify how to set the noise to simultaneously achieve good privacy and utility. As mentioned in Section 1, we will use the binary tree mechanism to set the $\mathbf{b}_t$'s.

It is worth mentioning that all three aspects of our algorithm are crucial: a) SRGs allow ensuring that the $\ell_2$-sensitivity of the gradient estimate at time $t$ is a factor of $1/t$, lower than that of the direct minibatch gradient ($\frac{1}{B} \sum_{d \in \mathcal{B}_t} \nabla f(\mathbf{x}_t; d)$), b) Noise addition via binary tree mechanism allows the standard deviation of the noise $\mathbf{b}_t$ to grow only polylogarithmically in $T$, as opposed to $\sqrt{T}$ via simple composition analysis (Dwork and Roth, 2014), and c) Accelerated SGD allows appropriately balancing the variance introduced by the SRG estimates, and that due to the i.i.d. sampling of the minibatches from the population. In fact in Section 4.3 we argue that it is necessary to have an accelerated SGD algorithm to achieve a single epoch optimal DP-SCO algorithm with batch size $B = \omega(1)$, if we were to use the SRGs as the gradient approximators.

---

**Algorithm 1** `Accelerated-DP-SRGD`

---

**Input:** Steps: $T$, batches: $\{\mathcal{B}_t\}_{0 \leq t < T}$ of size $B$, learning rates: $\{\eta_t\}_{0 \leq t < T}$, scaling $\beta$, interpolation params: $\{\tau_t\}_{t \in [T]}$, $\mathcal{C} : \|\mathcal{C}\|_2 \leq R$.

$\mathbf{x}_0, \mathbf{z}_0 \leftarrow 0$.

**for** $t \in \{0, \dots, T-1\}$ **do**

$\quad \Delta_t \leftarrow \frac{1}{B} \sum_{d \in \mathcal{B}_t} [\eta_t \nabla f(\mathbf{x}_t, d) - \eta_{t-1} \nabla f(\mathbf{x}_{t-1}, d)]$

$\quad \nabla_t \leftarrow \frac{\sum_{i \leq t} \Delta_i}{\eta_t}$ (equivalently, $\nabla_t \leftarrow \frac{\Delta_t}{\eta_t} + \frac{\eta_{t-1}}{\eta_t} \cdot \nabla_{t-1}$)

$\quad \mathbf{z}_{t+1} \leftarrow \Pi_{\mathcal{C}} \left( \mathbf{z}_t - \frac{\eta_t}{\beta} \nabla_t + \mathbf{b}_t \right)$, where $\mathbf{b}_t$ is the noise added to ensure privacy.

$\quad \mathbf{y}_{t+1} \leftarrow \Pi_{\mathcal{C}} \left( \mathbf{x}_t - \frac{1}{\beta} \nabla_t + \frac{1}{\eta_t} \cdot \mathbf{b}_t \right)$.

$\quad \mathbf{x}_{t+1} \leftarrow (1 - \tau_{t+1}) \mathbf{y}_{t+1} + \tau_{t+1} \mathbf{z}_{t+1}$.

**end**

---

## 4. Overview of Analysis of Algorithm 1

In this section we provide a high-level overview of the analysis of Algorithm 1.

### 4.1. Utility Analysis

First, ignoring privacy concerns we provide a excess population risk guarantee of Algorithm 1 for any choice of the privacy noise random variables $\{\mathbf{b}_t\}$'s. We only assume that the variance (over the sampling of the data samples $d \sim \mathcal{D}$) in any individual term in the SRGs (i.e., $\eta_t \nabla f(\mathbf{x}_t, d) - \eta_{t-1} \nabla f(\mathbf{x}_{t-1}, d)$) is bounded by $S^2$.

Eventually in Theorem 2 we will combine Theorem 1 below with the privacy analysis of Algorithm 1 to arrive at the final DP-SCO guarantee, with the appropriate choice of the batch gradient steps, and the batch size. In particular, there we instantiate the noise random variables $\{\mathbf{b}_t\}$'s with the (by now standard) *binary tree mechanism* (Chan et al., 2011; Dwork et al., 2010).

**Theorem 1 (Excess population risk; Simplification of Theorem 3)** *Fix some setting of the $\mathbf{b}_t$ such that $\|\mathbf{b}_t\|_2 \le b_{max}$ for all $t$. For an appropriate setting of $\eta_t, \tau_t$ and $\beta \ge M$, suppose $\mathbb{E}[\|\Delta_t - \mathbb{E}[\Delta_t]\|_2^2] \le S^2$. Then, with high probability over the randomness of the batches $\mathcal{B}$, the following is true for Algorithm 1:*

$$\mathbb{E}\left[F(\mathbf{y}_T)\right] - F(\mathbf{x}^*) =$$
$$\widetilde{O}\left(\frac{SL + S^2}{\beta\sqrt{T}} + \frac{SR}{T^{3/2}} + \frac{(S+L)b_{max}}{T} + \frac{\beta b_{max} R}{T^2} + \frac{\beta}{T^2}R^2\right).$$

**Interpretation of Theorem 1:** To understand the bound in Theorem 1, we can view it as three groups of terms. The first group is $\frac{SL+S^2}{\beta\sqrt{T}} + \frac{SR}{T^{3/2}}$, which is the error due to *stochasticity of the gradients*. The second is $\frac{(S+L)b_{max}}{T} + \frac{\beta b_{max} R}{T^2}$, which is the error due to *noise added for privacy*. The last is $\frac{\beta}{T^2}R^2$, which is the *optimization error*, i.e. the error of the algorithm without stochasticity or DP noise. Notice that the optimization error matches that of the non-stochastic part of non-private Nesterov's acceleration (Lan, 2012). Without privacy, the stochastic part of Nesterov's acceleration depends on $\frac{V}{\beta\sqrt{T}}$ (Lan, 2012), where $V$ bounds stochasticity in evaluating the population gradient:

$$\mathbb{E}_{\mathcal{B}_t}\left[\left\|\frac{1}{|\mathcal{B}_t|}\sum_{d\in\mathcal{B}_t}\nabla f(\mathbf{x};d) - \nabla F(\mathbf{x};d)\right\|_2^2\right] \le V^2.$$

Morally, the parameter $S$ in Theorem 1 can be compared to $V$. However, $S$ encodes the additional variance due to stochastic recursive gradients. Modulo the difference between $S$ and $V$, Theorem 1 cleanly decomposes the non-private stochastic Nesterov's utility and the additional error due to the explicit noise introduced due to DP.

**Proof overview:** We define the following potential. Let $\eta_{0:t} = \sum_{i=0}^{t} \eta_i$, then

$$\Phi_t = \eta_{0:t-1}\left(F(\mathbf{y}_t) - F(\mathbf{x}^*)\right) + 2\beta\|\mathbf{z}_t - \mathbf{x}^*\|_2^2.$$

The proof has three parts: a) **We bound the potential drop in each iteration** with terms arising from the stochasticity of sampling the $\{f_t\}$'s and the privacy noise $\{\mathbf{b}_t\}$'s, b) **Use telescopic summation of the potential drops** to bound the final potential $\Phi_T$ as a function of stochastic error and DP noise across all steps, and c) **convert the potential bound to a loss bound** to obtain the final error guarantee. Our proof closely follows the proof structure of Bansal and Gupta (2019) and draws inspiration from Xiao (2009) in terms of parameter choices.

**Independent batch gradients:** In Theorem 1 we analyzed Algorithm 1 where the $\nabla_t$ are computed using SRG. The analysis can easily be extended to the standard setting where each $\nabla_t$ is an independent batch gradient. We give this analysis which may be of independent interest in Section 5.5.

## 4.2. Privacy analysis and DP-SCO bounds

**Bounding gradient differences:** We next need to bound the norms of the gradient differences for a single example $\Delta_t(d) := \eta_t \nabla f(\mathbf{x}_t, d) - \eta_{t-1} \nabla f(\mathbf{x}_{t-1}, d)$. This gives us a bound on the variance $S^2$ in the loss bound of Theorem 1, and a bound on the sensitivity of the $\Delta_t$, which will let us bound the noise $\mathbf{b}_t$ needed for privacy.

By smoothness and Lipschitzness, for any $d$ we have:

$$\eta_t \nabla f(\mathbf{x}_t, d) - \eta_{t-1} \nabla f(\mathbf{x}_{t-1}, d) = \eta_t(\nabla f(\mathbf{x}_t, d) - \nabla f(\mathbf{x}_{t-1}, d)) + (\eta_t - \eta_{t-1}) \nabla f(\mathbf{x}_{t-1}, d)$$
$$\leq \eta_t M \|\mathbf{x}_t - \mathbf{x}_{t-1}\|_2 + (\eta_t - \eta_{t-1}) L.$$

We will choose $\eta_t = t + 1$, so the second term is just $L$. The main challenge is then to bound the first term. We will choose $\tau_t \approx 1/t$ to be vanishing, so

$$\mathbf{x}_t = (1 - \tau_t)\mathbf{y}_t + \tau_t \mathbf{z}_t \approx \mathbf{y}_t,$$

and in turn

$$\eta_t M \|\mathbf{x}_t - \mathbf{x}_{t-1}\|_2 \approx \eta_t M \|\mathbf{y}_t - \mathbf{x}_{t-1}\|_2 \leq \left\| \frac{\eta_t M}{\beta} \cdot \nabla_t + \frac{M}{\beta} \mathbf{b}_t \right\|_2.$$

The term $\frac{M}{\beta} \mathbf{b}_t$ ends up having norm at most $L$ under the right parameter choices, so the main challenge is to bound $\frac{\eta_t M}{\beta} \cdot \nabla_t$. Our approach for bounding this term differs based on whether or not we assume $\mathbf{x}^\dagger \in \mathcal{C}$. If $\mathbf{x}^\dagger \in \mathcal{C}$, $\mathbf{x}^* = \mathbf{x}^\dagger$ and by smoothness we have

$$\|\nabla F(\mathbf{x}_t)\|_2 \leq \sqrt{2M(F(\mathbf{x}_t) - F(\mathbf{x}^*))}.$$

Then since $\nabla_t$ is a stochastic estimate of $\nabla F(\mathbf{x}_t)$, by Theorem 1 and a concentration inequality on $\nabla_t$, the fact that $F(\mathbf{x}_t) - F(\mathbf{x}^*)$ implies a bound on $\|\nabla_t\|_2$ that is also decreasing. In turn, while $\eta_t$ is increasing, we are able to show $\frac{\eta_t M}{\beta} \cdot \nabla_t$ remains constant even with $\beta = O(M)$ (Lemma 7). A subtle challenge is that in order to obtain a bound on $\|\nabla_t\|_2$ that is decreasing, we actually employ a tighter version of Theorem 1 that uses the exact norms $\|\nabla F(\mathbf{x}_t)\|_2$ rather than the Lipschitz constant $L$ in the final bound. In turn, there is a circular dependence between the bound of Theorem 1 and the norm of the gradient differences. Because of the circular dependence, bounding both simultaneously requires a proof by induction with careful choice of parameters (Lemma 6).

Without assuming $\mathbf{x}^\dagger \in \mathcal{C}$, it's no longer the case that $\nabla_t$ is necessarily decreasing over time (e.g. in the case of linear losses, $\nabla_t$ will remain constant). In this case, we more directly bound $\frac{\eta_t M}{\beta} \cdot \nabla_t$ by choosing $\beta \geq TM \geq \eta_t M$, which ensures the gradient differences still have constant sensitivity (Lemma 11). The fact that we have to increase $\beta$ (i.e., slow down learning) means we need to increase $T$ to compensate in utility, leading to the increased iteration complexity $T = \sqrt{n}$ instead of $T \approx n^{1/4}$.

**Privatizing the gradient differences:** Once we have a bound on $\Delta_t(d)$, we now need to bound $\max_t \|\mathbf{b}_t\|_2$, i.e. the maximum noise added to any gradient estimate $\nabla_t$. Since each $\nabla_t$ is a (rescaling of) prefix sums of $\Delta_t$, we do this using the binary tree mechanism of Dwork et al. (2010), which adds noise to the prefix sums with variance scaling as $\text{polylog}\,(T)$ rather than $T$, what we would get by independently noising each $\Delta_t$. We can use existing results on the binary tree mechanism (for convenience, reproven as Lemma 8) to obtain bounds on the maximum noise $\max_t \|\mathbf{b}_t\|_2$ needed to achieve $(\varepsilon, \delta)$-DP.

Finally, it is straightforward to get our main results by plugging in the bounds on $\max_t \|\mathbf{b}_t\|_2$ and $\Delta_t(d)$ into Theorem 1:

**Theorem 2 (Simplified version of Theorems 10 and 14)** *For appropriate settings of $\beta, T$, Algorithm 1 is $(\varepsilon, \delta)$-DP and satisfies:*

$$F(\mathbf{y}_T) - F(\mathbf{x}^*) = O\left( LR \cdot \text{polylog}\,(n, 1/\delta) \cdot \left( \frac{1}{\sqrt{n}} + \frac{\sqrt{p}}{\varepsilon n} \right) \right).$$

*If we assume $\mathbf{x}^\dagger \in \mathcal{C}$, then the parameter settings are such that $T = \widetilde{\Theta}(n^{1/4})$ and $BT = n$, i.e. only a single epoch is required. Without this assumption, the parameter settings are such that we can allow any $T \geq \sqrt{n}$ and $BT = n$, i.e. still only a single epoch is required and the batch size can be as large as $\sqrt{n}$.*

### 4.3. Unaccelerated Variant

A natural question is whether an unaccelerated variant of `Accelerated-DP-SRGD` can also achieve optimal DP-SCO rates with batch size $\omega(1)$. We give a sketch that the unaccelerated version requires batch size $O(1)$. Suppose we use $\Delta_t = \eta_t \nabla f(\mathbf{x}_t, \mathcal{B}_t) - \eta_{t-1} \nabla f(\mathbf{x}_{t-1}, \mathcal{B}_t)$ as our gradient differences and $\nabla_t = \frac{\sum_{i \leq t} \Delta_i}{\eta_t}$ as our stochastic recursive gradients. For simplicity we will consider using SGD with a fixed learning rate $1/\beta$ as the underlying optimization algorithm, assume $L = R = M = 1$ and assume $\eta_t$ grows at most polynomially, which implies $\sum_{i \leq t} \eta_i^2 \approx t\eta_t^2$. Similarly to our sensitivity bound for the accelerated algorithm (Lemma 11), we can show the sensitivity of each $\Delta_t$ is roughly $\eta_t/\beta$. Suppose we use $1/\beta \approx \sqrt{B/T} = B/\sqrt{n}$, which is standard for interpolation between full-batch GD and SGD with $B = 1$ (see e.g. Section 6 of Bubeck (2015)). We compute the vector variance $\mathbf{Var}\,(\mathbf{v}) := \mathbb{E}[\|\mathbf{v} - \mathbb{E}[\mathbf{v}]\|_2^2]$ of $\nabla_t$:

$$\mathbf{Var}\,(\nabla_t) = \mathbf{Var}\left( \frac{\sum_{i \leq t} \Delta_t}{\eta_t} \right) = \frac{\sum_{i \leq t} \mathbf{Var}\,(\Delta_t)}{\eta_t^2} = \frac{\sum_{i \leq t} \eta_i^2}{\eta_t^2 \beta^2} = \frac{t}{B\beta^2} = \frac{tB}{n}.$$

For $t = \Omega(T) = \Omega(n/B)$, the variance is $\Omega(1)$. In other words, regardless of the batch size, the variance of the gradients for the majority of iterations is comparable to that of vanilla SGD with batch size 1. SGD with batch size 1, i.e. with variance $O(1)$ rather than $O(1/B)$, requires $T = n$ iterations to converge, hence in the single-epoch setting we need $B = O(1)$.

## 5. Utility and Privacy Analysis

In this section, we provide the detailed utility analysis and privacy analysis of Algorithm 1.

### 5.1. Utility Analysis

First, we provide the excess population risk guarantee for any choice of the privacy noise random variables $\{\mathbf{b}_t\}$'s. We only assume that the variance (over the sampling of the data samples $d \sim \mathcal{D}$) in any individual term in the SRGs (i.e., $\eta_t \nabla f(\mathbf{x}_t, d) - \eta_{t-1} \nabla f(\mathbf{x}_{t-1}, d)$) is bounded.

**Theorem 3 (Excess population risk)** *Fix some setting of the $\mathbf{b}_t$. Let $\eta_t, \beta$ be such that $\eta_t$ is non-decreasing, $\eta_t^2 \leq 4\eta_{0:t}$, and $\beta \geq M$. For any $\mathbf{x}_t, \mathbf{x}_{t-1}$, and for all $d \in \mathcal{D}$, assume*

$$\|(\eta_t \nabla f(\mathbf{x}_t, d) - \eta_{t-1} \nabla f(\mathbf{x}_{t-1}, d)) - \mathbb{E}_d[\eta_t \nabla f(\mathbf{x}_t, d) - \eta_{t-1} \nabla f(\mathbf{x}_{t-1}, d)]\|_2 \leq Q.$$

*Then, w.p. $1 - \gamma$ over the randomness of the i.i.d. sampling of the batches $\{\mathcal{B}_t\}$, the following is true for Algorithm 1:*

$$F(\mathbf{y}_T) - F(\mathbf{x}^*) \leq$$

$$\frac{1}{\eta_{0:T}} \sum_{t=0}^{T-1} \eta_{0:t} \left( \frac{2Q\sqrt{3(t+1)\log(6T/\gamma)}(\|\nabla F(\mathbf{x}_t)\|_2 + \frac{Q\sqrt{3(t+1)\log(6T/\gamma)}}{\sqrt{B}\eta_t})}{\sqrt{B}\eta_t\beta} \right.$$

$$+ \frac{3Q\sqrt{3(t+1)\log(6T/\gamma)}\|\mathbf{b}_t\|_2}{\sqrt{B}\eta_t^2} + \frac{Q\|\mathcal{C}\|_2\sqrt{3(t+1)\log(6T/\gamma)}}{\sqrt{B}\eta_{0:t}} + \frac{3\beta}{\eta_{0:t}}\|\mathbf{b}_t\|_2 \cdot \|\mathcal{C}\|_2$$

$$\left. + \frac{\beta\|\mathbf{b}_t\|_2^2}{\eta_t^2} \right) + \frac{\beta}{\eta_{0:T}}\|\mathcal{C}\|_2^2.$$

**Proof**

**Potential drop:** We define

$$\Phi_t = \eta_{0:t-1}(F(\mathbf{y}_t) - F(\mathbf{x}^*)) + 2\beta\|\mathbf{z}_t - \mathbf{x}^*\|_2^2. \tag{1}$$

We first show that the growth in the potential $\Phi_t$ at each iteration of Algorithm 1 is bounded. In particular, if the stochasticity of the algorithm is removed, i.e., if we set $Q = 0$ and $\{\mathbf{b}_t\}$'s to zero, then the potential growth is non-positive. (This is consistent with the full-gradient version of Nesterov's acceleration (Bansal and Gupta, 2019).)

**Lemma 4 (Potential drop)** *Let $\mathbf{q}_t = \nabla_t - \nabla F(\mathbf{x}_t)$. Assuming $\eta_t^2 \leq 4\eta_{0:t}$, w.p. $1 - \gamma$ over $\{\mathcal{B}_t\}$ the potential drop is bounded by the following (where $\Delta\Phi_t = \Phi_{t+1} - \Phi_t$):*

$$\Delta\Phi_t \le \eta_{0:t}\left(\frac{2Q\sqrt{3(t+1)\log(6/\gamma)}(\|\nabla F(\mathbf{x}_t)\|_2 + \frac{Q\sqrt{3(t+1)\log(6/\gamma)}}{\sqrt{B}\eta_t})}{\sqrt{B}\eta_t\beta}\right.$$

$$+ \frac{3Q\sqrt{3(t+1)\log(6/\gamma)}\|\mathbf{b}_t\|_2}{\sqrt{B}\eta_t^2} + \frac{Q\|\mathcal{C}\|_2\sqrt{3(t+1)\log(6/\gamma)}}{\sqrt{B}\eta_{0:t}}$$

$$\left.+ \frac{3\beta}{\eta_{0:t}}\|\mathbf{b}_t\|_2\cdot\|\mathcal{C}\|_2 + \frac{\beta\|\mathbf{b}_t\|_2^2}{\eta_t^2}\right).$$

**Telescopic summation and averaging:** By telescoping the potential differences $\Delta\Phi_t$'s and a union bound over failure probabilities we have the following w.p. $1-\gamma$ over the sampling of the functions $\{f_t\}$'s:

$$\Phi_T - \Phi_0 \le \sum_{t=0}^{T-1}\eta_{0:t}\left(\frac{2Q\sqrt{3(t+1)\log(6T/\gamma)}(\|\nabla F(\mathbf{x}_t)\|_2 + \frac{Q\sqrt{3(t+1)\log(6T/\gamma)}}{\sqrt{B}\eta_t})}{\sqrt{B}\eta_t\beta}\right.$$

$$+ \frac{3Q\sqrt{3(t+1)\log(6T/\gamma)}\|\mathbf{b}_t\|_2}{\sqrt{B}\eta_t^2} + \frac{Q\|\mathcal{C}\|_2\sqrt{3(t+1)\log(6T/\gamma)}}{\sqrt{B}\eta_{0:t}} \tag{2}$$

$$\left.+ \frac{3\beta}{\eta_{0:t}}\|\mathbf{b}_t\|_2\cdot\|\mathcal{C}\|_2 + \frac{\beta\|\mathbf{b}_t\|_2^2}{\eta_t^2}\right). \tag{3}$$

Therefore, from Equation (3) and definition of $\Phi_t$, as desired we have w.p. $1-\gamma$:

$$F(\mathbf{y}_T) - F(\mathbf{x}^*) \le$$

$$\frac{1}{\eta_{0:T}}\sum_{t=0}^{T-1}\eta_{0:t}\left(\frac{2Q\sqrt{3(t+1)\log(6T/\gamma)}(\|\nabla F(\mathbf{x}_t)\|_2 + \frac{Q\sqrt{3(t+1)\log(6T/\gamma)}}{\sqrt{B}\eta_t})}{\sqrt{B}\eta_t\beta}\right.$$

$$+ \frac{3Q\sqrt{3(t+1)\log(6T/\gamma)}\|\mathbf{b}_t\|_2}{\sqrt{B}\eta_t^2} + \frac{Q\|\mathcal{C}\|_2\sqrt{3(t+1)\log(6T/\gamma)}}{\sqrt{B}\eta_{0:t}} + \frac{3\beta}{\eta_{0:t}}\|\mathbf{b}_t\|_2\cdot\|\mathcal{C}\|_2$$

$$\left.+ \frac{\beta\|\mathbf{b}_t\|_2^2}{\eta_t^2}\right) + \frac{\beta}{\eta_{0:T}}\|\mathcal{C}\|_2^2.$$

This completes the proof of Theorem 3. ∎

**Proof** [Proof of Lemma 4] We have

$$\Delta\Phi_t = \underbrace{\eta_{0:t}(F(\mathbf{y}_{t+1}) - F(\mathbf{x}_t))}_{A} - \underbrace{\eta_{0:t-1}(F(\mathbf{y}_t) - F(\mathbf{x}_t)) + \eta_t(F(\mathbf{x}_t) - F(\mathbf{x}^*))}_{B}$$

$$+ \underbrace{2\beta\left(\|\mathbf{z}_{t+1} - \mathbf{x}^*\|_2^2 - \|\mathbf{z}_t - \mathbf{x}^*\|_2^2\right)}_{C}. \tag{4}$$

By convexity, we have the following linearization bound on term $B$ in Equation (4):

$$B \leq -\eta_{0:t-1} \langle \nabla F(\mathbf{x}_t), \mathbf{y}_t - \mathbf{x}_t \rangle + \eta_t \langle \nabla F(\mathbf{x}_t), \mathbf{x}_t - \mathbf{x}^* \rangle \tag{5}$$

Now, $(1 - \tau_t)(\mathbf{y}_t - \mathbf{x}_t) = \tau_t(\mathbf{x}_t - \mathbf{z}_t)$. Setting, $\tau_t = \frac{\eta_t}{\eta_{0:t}}$ in Equation (5) (and using the definition of $\mathbf{q}_t$), we have the following:

$$B \leq \eta_t \langle \nabla F(\mathbf{x}_t), \mathbf{z}_t - \mathbf{x}^* \rangle = \eta_t \langle \nabla_t, \mathbf{z}_t - \mathbf{x}^* \rangle - \eta_t \langle \mathbf{q}_t, \mathbf{z}_t - \mathbf{x}^* \rangle . \tag{6}$$

Now, we bound the term $C$ in Equation (4) as follows. Let $\mathbf{z}'_{t+1} = \mathbf{z}_t - \frac{\eta_t}{2\beta} \nabla_t + \mathbf{b}_t$.

$$
\begin{aligned}
C &= 2\beta \left( \|\mathbf{z}_{t+1} - \mathbf{x}^*\|_2^2 - \|\mathbf{z}_t - \mathbf{x}^*\|_2^2 \right) \\
&= 2\beta \left( 2 \langle \mathbf{z}_{t+1} - \mathbf{z}_t, \mathbf{z}_{t+1} - \mathbf{x}^* \rangle - \|\mathbf{z}_{t+1} - \mathbf{z}_t\|_2^2 \right) \\
&\leq 2\beta \left( \left\langle -\frac{\eta_t}{\beta} \nabla_t + \mathbf{b}_t, \mathbf{z}_{t+1} - \mathbf{x}^* \right\rangle - \|\mathbf{z}_{t+1} - \mathbf{z}_t\|_2^2 \right) \tag{7} \\
&= -2\beta \left( \left\langle \frac{\eta_t}{\beta} \nabla_t, \mathbf{z}_{t+1} - \mathbf{x}^* \right\rangle + \|\mathbf{z}_{t+1} - \mathbf{z}_t\|_2^2 \right) + 2\beta \langle \mathbf{b}_t, \mathbf{z}_{t+1} - \mathbf{x}^* \rangle . \tag{8}
\end{aligned}
$$

In Equation (7), we used the Pythagorean property of convex projection. Combining Equation (8) and Equation (6), we have the following:

$$B + C \leq \eta_t \langle \nabla_t, \mathbf{z}_t - \mathbf{z}_{t+1} \rangle - 2\beta \|\mathbf{z}_{t+1} - \mathbf{z}_t\|_2^2 + 2\beta \langle \mathbf{b}_t, \mathbf{z}_{t+1} - \mathbf{x}^* \rangle - \eta_t \langle \mathbf{q}_t, \mathbf{z}_t - \mathbf{x}^* \rangle \tag{9}$$

Using Lemma 5 below, and the fact that $\tau_t \leq 1$, we have the potential drop in Lemma 4 bound by the following.

$$\Delta \Phi_t \leq -\eta_{0:t} \langle \mathbf{q}_t, \mathbf{y}_{t+1} - \mathbf{x}_t \rangle + \frac{L \|\mathbf{b}_t\|_2}{\tau_t} + \frac{\beta \|\mathbf{b}_t\|_2^2}{\eta_t \tau_t} + 3\beta \|\mathbf{b}_t\|_2 \|\mathcal{C}\|_2 - \eta_t \langle \mathbf{q}_t, \mathbf{z}_t - \mathbf{x}^* \rangle \tag{10}$$

Recall, by definition $\mathbf{y}_{t+1} = \underset{\mathbf{y} \in \mathcal{C}}{\arg\min} \left\langle \mathbf{q}_t + \nabla F(\mathbf{x}_t) - \frac{\beta}{\eta_t} \cdot \mathbf{b}_t, \mathbf{y} - \mathbf{x}_t \right\rangle + \frac{\beta}{2} \|\mathbf{y} - \mathbf{x}_t\|_2^2$. Let $\mathbf{y}'_{t+1} = \mathbf{x}_t - \frac{1}{\beta} \nabla_t + \frac{1}{\eta_t} \cdot \mathbf{b}_t$ be the unconstrained minimizer, and let

$$J(\mathbf{y}) = \left\langle \mathbf{q}_t + \nabla F(\mathbf{x}_t) - \frac{\beta}{\eta_t} \cdot \mathbf{b}_t, \mathbf{y} \right\rangle + \frac{\beta}{2} \|\mathbf{y}\|_2^2 .$$

By definition, $J(\mathbf{y}_{t+1}) \geq J(\mathbf{y}'_{t+1}) + \frac{\beta}{2} \|\mathbf{y}_{t+1} - \mathbf{y}'_{t+1}\|_2^2$. Therefore,

$$
\begin{aligned}
\left\langle \mathbf{q}_t + \nabla F(\mathbf{x}_t) - \frac{\beta}{\eta_t} \cdot \mathbf{b}_t, \mathbf{y}_{t+1} - \mathbf{y}'_{t+1} \right\rangle &\geq \frac{\beta}{2} \|\mathbf{y}_{t+1} - \mathbf{y}'_{t+1}\|_2^2 \\
\Rightarrow \|\mathbf{y}_{t+1} - \mathbf{y}'_{t+1}\|_2 &\leq \frac{2 \|\nabla_t\|_2}{\beta} + \frac{2 \|\mathbf{b}_t\|_2}{\eta_t} . \tag{11}
\end{aligned}
$$

We now apply (11) and Cauchy-Schwarz inequality to (10):

$$\Delta\Phi_t \leq -\eta_{0:t}\langle \mathbf{q}_t, \mathbf{y}_{t+1} - \mathbf{x}_t\rangle + \frac{L\|\mathbf{b}_t\|_2}{\tau_t} + \frac{\beta\|\mathbf{b}_t\|_2^2}{\eta_t\tau_t} + 3\beta\|\mathbf{b}_t\|_2\|\mathcal{C}\|_2 - \eta_t\langle \mathbf{q}_t, \mathbf{z}_t - \mathbf{x}^*\rangle$$

$$= -\eta_{0:t}\langle \mathbf{q}_t, \mathbf{y}_{t+1} - \mathbf{y}'_{t+1}\rangle - \eta_{0:t}\langle \mathbf{q}_t, \mathbf{y}'_{t+1} - \mathbf{x}_t\rangle + \frac{L\|\mathbf{b}_t\|_2}{\tau_t} + \frac{\beta\|\mathbf{b}_t\|_2^2}{\eta_t\tau_t}$$

$$+ 3\beta\|\mathbf{b}_t\|_2\|\mathcal{C}\|_2 - \eta_t\langle \mathbf{q}_t, \mathbf{z}_t - \mathbf{x}^*\rangle$$

$$= -\eta_{0:t}\langle \mathbf{q}_t, \mathbf{y}_{t+1} - \mathbf{y}'_{t+1}\rangle - \eta_{0:t}\left\langle \mathbf{q}_t, -\frac{1}{\beta}\nabla_t + \frac{1}{\eta_t}\mathbf{b}_t\right\rangle + \frac{L\|\mathbf{b}_t\|_2}{\tau_t} + \frac{\beta\|\mathbf{b}_t\|_2^2}{\eta_t\tau_t}$$

$$+ 3\beta\|\mathbf{b}_t\|_2\|\mathcal{C}\|_2 - \eta_t\langle \mathbf{q}_t, \mathbf{z}_t - \mathbf{x}^*\rangle$$

$$\leq \eta_{0:t}\|\mathbf{q}_t\|_2\|\mathbf{y}_{t+1} - \mathbf{y}'_{t+1}\|_2 + \frac{\eta_{0:t}}{\beta}\|\mathbf{q}_t\|_2\|\nabla_t\|_2 + \frac{\eta_{0:t}}{\eta_t}\|\mathbf{q}_t\|_2\|\mathbf{b}_t\|_2 + \frac{L\|\mathbf{b}_t\|_2}{\tau_t} + \frac{\beta\|\mathbf{b}_t\|_2^2}{\eta_t\tau_t}$$

$$+ 3\beta\|\mathbf{b}_t\|_2\|\mathcal{C}\|_2 + \eta_t\|\mathbf{q}_t\|_2\|\mathbf{z}_t - \mathbf{x}^*\|_2$$

$$\leq \eta_{0:t}\|\mathbf{q}_t\|_2\|\mathbf{y}_{t+1} - \mathbf{y}'_{t+1}\|_2 + \frac{\eta_{0:t}}{\beta}\|\mathbf{q}_t\|_2\|\nabla_t\|_2 + \frac{\eta_{0:t}}{\eta_t}\|\mathbf{q}_t\|_2\|\mathbf{b}_t\|_2 + \frac{L\|\mathbf{b}_t\|_2}{\tau_t} + \frac{\beta\|\mathbf{b}_t\|_2^2}{\eta_t\tau_t}$$

$$+ 3\beta\|\mathbf{b}_t\|_2\|\mathcal{C}\|_2 + \eta_t\|\mathbf{q}_t\|_2\|\mathcal{C}\|_2$$

$$\leq \frac{3\eta_{0:t}}{\beta}\|\mathbf{q}_t\|_2\|\nabla_t\|_2 + \frac{3\eta_{0:t}}{\eta_t}\|\mathbf{q}_t\|_2\|\mathbf{b}_t\|_2 + \frac{L\|\mathbf{b}_t\|_2}{\tau_t} + \frac{\beta\|\mathbf{b}_t\|_2^2}{\eta_t\tau_t}$$

$$+ 3\beta\|\mathbf{b}_t\|_2\|\mathcal{C}\|_2 + \eta_t\|\mathbf{q}_t\|_2\|\mathcal{C}\|_2. \tag{12}$$

Now, we observe that:

$$\mathbf{q}_t = \nabla_t - \nabla F(\mathbf{x}_t) = \frac{\sum_{i\leq t}\sum_{d\in\mathcal{B}_i} \begin{array}{c}((\eta_t\nabla f(\mathbf{x}_t, d) - \eta_{t-1}\nabla f(\mathbf{x}_{t-1}, d)) \\ - \mathbb{E}_d[\eta_t\nabla f(\mathbf{x}_t, d) - \eta_{t-1}\nabla f(\mathbf{x}_{t-1}, d)])\end{array}}{B\eta_t}.$$

The sum $\sum_{i\leq t}\sum_{d\in\mathcal{B}_i}((\eta_t\nabla f(\mathbf{x}_t, d) - \eta_{t-1}\nabla f(\mathbf{x}_{t-1}, d)) - \mathbb{E}_d[\eta_t\nabla f(\mathbf{x}_t, d) - \eta_{t-1}\nabla f(\mathbf{x}_{t-1}, d)])$ is a mean-zero vector martingale with $B(t+1)$ terms each with norm at most $Q$ with probability 1 by the assumptions of the theorem. By concentration of bounded vector martingales (e.g., (Hayes, 2005)), w.p. $1 - \gamma$ we have

$$\left\|\sum_{i\leq t}\sum_{d\in\mathcal{B}_i}\begin{array}{c}((\eta_t\nabla f(\mathbf{x}_t, d) - \eta_{t-1}\nabla f(\mathbf{x}_{t-1}, d)) \\ - \mathbb{E}_d[\eta_t\nabla f(\mathbf{x}_t, d) - \eta_{t-1}\nabla f(\mathbf{x}_{t-1}, d)])\end{array}\right\|_2^2 \leq 3Q^2 B(t+1)\log(6/\gamma).$$

and thus $\|\mathbf{q}_t\|_2 \leq \frac{Q\sqrt{3(t+1)\log(6/\gamma)}}{\sqrt{B}\eta_t}$ with the same probability. Under this event we also have $\|\nabla_t\|_2 \leq \|\nabla F(\mathbf{x}_t)\|_2 + \frac{Q\sqrt{3(t+1)\log(6/\gamma)}}{\sqrt{B}\eta_t}$ by triangle inequality. Plugging these bounds into (12) we have w.p. $1 - \gamma$:

$$\Delta\Phi_t \leq \eta_{0:t} \left( \frac{2Q\sqrt{3(t+1)\log(6/\gamma)}(\|\nabla F(\mathbf{x}_t)\|_2 + \frac{Q\sqrt{3(t+1)\log(6/\gamma)}}{\sqrt{B}\eta_t})}{\sqrt{B}\eta_t\beta} \right.$$

$$+ \frac{3Q\sqrt{3(t+1)\log(6/\gamma)}\|\mathbf{b}_t\|_2}{\sqrt{B}\eta_t^2} + \frac{Q\|\mathcal{C}\|_2\sqrt{3(t+1)\log(6/\gamma)}}{\sqrt{B}\eta_{0:t}}$$

$$\left. + \frac{3\beta}{\eta_{0:t}}\|\mathbf{b}_t\|_2 \cdot \|\mathcal{C}\|_2 + \frac{\beta\|\mathbf{b}_t\|_2^2}{\eta_t^2} \right).$$

∎

**Lemma 5** *Assuming $\eta_t^2 \leq 4\eta_{0:t}$. We have the following:*

$$\eta_{0:t}(F(\mathbf{y}_{t+1}) - F(\mathbf{x}_t)) + \eta_t\langle\nabla_t, \mathbf{z}_t - \mathbf{z}_{t+1}\rangle - 2\beta\|\mathbf{z}_{t+1} - \mathbf{z}_t\|_2^2 \leq$$

$$-\eta_{0:t}\langle\mathbf{q}_t, \mathbf{y}_{t+1} - \mathbf{x}_t\rangle + \left( \frac{L\|\mathbf{b}_t\|_2}{\tau_t} + \frac{\beta\|\mathbf{b}_t\|_2^2}{\eta_t\tau_t} + \beta\|\mathbf{b}_t\|_2\|\mathcal{C}\|_2 \right).$$

**Proof** [Proof of Lemma 5] By smoothness we have,

$$F(\mathbf{y}_{t+1}) - F(\mathbf{x}_t) \leq \langle\nabla_t, \mathbf{y}_{t+1} - \mathbf{x}_t\rangle - \langle\mathbf{q}_t, \mathbf{y}_{t+1} - \mathbf{x}_t\rangle + \frac{M}{2}\|\mathbf{y}_{t+1} - \mathbf{x}_t\|_2^2 \tag{13}$$

By definition $\mathbf{y}_{t+1} = \underset{\mathbf{y}\in\mathcal{C}}{\arg\min}\left(\langle\nabla_t - \frac{\beta}{\eta_t}\cdot\mathbf{b}_t, \mathbf{y} - \mathbf{x}_t\rangle + \frac{\beta}{2}\|\mathbf{y} - \mathbf{x}_t\|_2^2\right)$. Therefore, for any $\mathbf{v}\in\mathcal{C}$, the following is true:

$$\left\langle\nabla_t - \frac{\beta}{\eta_t}\cdot\mathbf{b}_t, \mathbf{y}_{t+1} - \mathbf{x}_t\right\rangle + \frac{\beta}{2}\|\mathbf{y}_{t+1} - \mathbf{x}_t\|_2^2 \leq \left\langle\nabla_t - \frac{\beta}{\eta_t}\cdot\mathbf{b}_t, \mathbf{v} - \mathbf{x}_t\right\rangle + \frac{\beta}{2}\|\mathbf{v} - \mathbf{x_t}\|_2^2$$

$$\Leftrightarrow \langle\nabla_t, \mathbf{y}_{t+1} - \mathbf{x}_t\rangle \leq \langle\nabla_t, \mathbf{v} - \mathbf{x}_t\rangle + \frac{\beta}{\eta_t}\cdot\langle\mathbf{b}_t, \mathbf{y}_{t+1} - \mathbf{v}\rangle + \frac{\beta}{2}\|\mathbf{v} - \mathbf{x_t}\|_2^2 - \frac{\beta}{2}\|\mathbf{y}_{t+1} - \mathbf{x_t}\|_2^2.$$
$$\tag{14}$$

Plugging Equation (14) into Equation (13) and using $\beta \geq M$ we have the following:

$$F(\mathbf{y}_{t+1}) - F(\mathbf{x}_t) \leq \langle\nabla_t, \mathbf{v} - \mathbf{x}_t\rangle - \langle\mathbf{q}_t, \mathbf{y}_{t+1} - \mathbf{x}_t\rangle + \frac{\beta}{\eta_t}\cdot\langle\mathbf{b}_t, \mathbf{y}_{t+1} - \mathbf{v}\rangle$$

$$-\frac{\beta}{2}\|\mathbf{y}_{t+1} - \mathbf{x}_t\|_2^2 + \frac{\beta}{2}\|\mathbf{v} - \mathbf{x}_t\|_2^2. \tag{15}$$

Define $\mathbf{v} = (1 - \tau_t)\mathbf{y}_t + \tau_t\mathbf{z}_{t+1}$. Which implies $\mathbf{v} - \mathbf{x}_t = \tau_t(\mathbf{z}_{t+1} - \mathbf{z}_t)$. Substituting $\mathbf{v}$ in Equation (15), we have the following:

$$F(\mathbf{y}_{t+1}) - F(\mathbf{x}_t) \leq \tau_t\langle\nabla_t, \mathbf{z}_{t+1} - \mathbf{z}_t\rangle - \langle\mathbf{q}_t, \mathbf{y}_{t+1} - \mathbf{x}_t\rangle + \frac{\beta}{\eta_t}\cdot\langle\mathbf{b}_t, \mathbf{y}_{t+1} - \mathbf{v}\rangle$$

$$-\frac{\beta}{2}\|\mathbf{y}_{t+1} - \mathbf{x}_t\|_2^2 + \frac{\beta\tau_t^2}{2}\|\mathbf{z}_{t+1} - \mathbf{z}_t\|_2^2. \tag{16}$$

Assume, $\frac{\eta_t^2}{\eta_{0:t}} \leq 4$. Combining Equation (16) and Equation (13), and setting $\tau_t = \frac{\eta_t}{\eta_{0:t}}$, we have the following:

$$\eta_{0:t}(F(\mathbf{y}_{t+1}) - F(\mathbf{x}_t)) + \eta_t \langle \nabla_t, \mathbf{z}_t - \mathbf{z}_{t+1} \rangle - 2\beta \|\mathbf{z}_{t+1} - \mathbf{z}_t\|_2^2 \leq \tag{17}$$

$$- \eta_{0:t} \langle \mathbf{q}_t, \mathbf{y}_{t+1} - \mathbf{x}_t \rangle + \frac{\beta}{\tau_t} \langle \mathbf{b}_t, \mathbf{y}_{t+1} - \mathbf{v} \rangle + \frac{\beta}{2} \left( \eta_{0:t} \cdot \tau_t^2 - 4 \right) \|\mathbf{z}_{t+1} - \mathbf{z}_t\|_2^2 - \frac{\beta \eta_{0:t}}{2} \|\mathbf{y}_{t+1} - \mathbf{x}_t\|_2^2$$

$$\leq - \eta_{0:t} \langle \mathbf{q}_t, \mathbf{y}_{t+1} - \mathbf{x}_t \rangle + \frac{\beta}{\tau_t} \|\mathbf{b}_t\|_2 \|\mathbf{y}_{t+1} - \mathbf{x}_t - \tau_t(\mathbf{z}_{t+1} - \mathbf{z}_t)\|_2$$

$$\leq - \eta_{0:t} \langle \mathbf{q}_t, \mathbf{y}_{t+1} - \mathbf{x}_t \rangle + \frac{\beta}{\tau_t} \|\mathbf{b}_t\|_2 \left( \frac{L}{\beta} + \frac{\|\mathbf{b}_t\|_2}{\eta_t} + \tau_t \|\mathcal{C}\|_2 \right)$$

$$\leq - \eta_{0:t} \langle \mathbf{q}_t, \mathbf{y}_{t+1} - \mathbf{x}_t \rangle + \left( \frac{L \|\mathbf{b}_t\|_2}{\tau_t} + \frac{\beta \|\mathbf{b}_t\|_2^2}{\eta_t \tau_t} + \beta \|\mathbf{b}_t\|_2 \|\mathcal{C}\|_2 \right). \tag{18}$$

This completes the proof of Lemma 5. ■

## 5.2. Privacy analysis

Given Theorem 1, our goal now is to bound $S$, which is equivalent to bounding the sensitivity of the $\Delta_t$, and then to show that there is a noise distribution for which $b_{max}$ is sufficiently small (with high probability) and we achieve $(\varepsilon, \delta)$-DP.

Before proceeding with the proofs, we will specify the setting of parameters we will use in advance:

$$\eta_t = t + 1,$$
$$c_{DP} = 4\sqrt{2} \log^{3/2}(n) \sqrt{\ln(2n/\delta)} \ln(2.5/\delta),$$
$$\beta = \frac{c_{DP}^2 L \max\{1, \sqrt{p}/\varepsilon\sqrt{n}\}}{R},$$
$$T = c_{DP} \cdot n^{1/4},$$
$$B = n/T = n^{3/4}/c_{DP}. \tag{19}$$

### 5.2.1. BOUNDING GRADIENT NORMS

We first need to bound the gradient norms in each round:

**Lemma 6** *Using the parameter choices in Eq. (19), additionally assume:*
- *Letting $\Delta_t(d) := \eta_t \nabla f(\mathbf{x}_t, d) - \eta_{t-1} \nabla f(\mathbf{x}_{t-1}, d)$, for all $t \leq T - 1$ and $d$, $\|\Delta_t(d)\|_2 \leq c_{DP} L$.*
- *For all $t$: $\|\mathbf{b}_t\|_2 \leq b_{max} := \frac{c_{DP}^2 L \sqrt{p}}{\varepsilon B \beta}$ (which under Eq. (19), is at most $Rc_{DP}/n^{1/4} \leq R$ for all sufficiently large $n$ assuming $\delta = 2^{-o(n^{1/6})}$).*

*Then, for any $t$, all sufficiently large $n$ and $\delta < 1/n$, and some constant $c_1$ we have under the probability $1 - \gamma$ events of Theorem 3:*

$$\|\nabla F(\mathbf{x}_t)\|_2 \le c_1 \sqrt{M} \left( \sqrt{\frac{c_{DP}LR \log(T/\gamma)}{\sqrt{B(t+1)}}} + \sqrt{\frac{\beta b_{max}R}{t+1}} + \frac{2\sqrt{\beta}R}{t+1} \right).$$

**Proof**

We proceed by induction. For all sufficiently large $n$, we have $c_{DP}^2 \ge c_M$ and so $\beta \ge c_{DP}^2 L/R \ge M$. So for $t = 0$ we have by smoothness and the assumption that $\mathbf{x}^* = \mathbf{x}^\dagger$, i.e. $\|\nabla F(\mathbf{x}^*)\|_2 = 0$:

$$\|\nabla F(\mathbf{x}_0)\|_2 \le M \|\mathbf{x}_0 - \mathbf{x}^*\|_2 \le MR \le \sqrt{M\beta}R.$$

Hence the lemma holds for $t = 0$ as long as $c_1 \ge 1/2$.

Now inductively assume the lemma holds for rounds $0, 1, \ldots, t-1$. We can plug in $Q = 2c_{DP}L$ into Theorem 3, and then using Eq. (19) and the assumptions of the lemma we can combine and simplify terms to show that for some constant $c_2$:

$$F(\mathbf{y}_t) - F(\mathbf{x}^*) \le c_2 \left( \frac{1}{t^2} \sum_{i=0}^{t-1} \frac{c_{DP}Li^{3/2}\sqrt{\log(T/\gamma)} \|\nabla F(\mathbf{x}_i)\|_2}{\sqrt{B}\beta} + \frac{c_{DP}LR \log(T/\gamma)}{\sqrt{B(t+1)}} \right.$$
$$\left. + \frac{\beta b_{max}R}{t+1} + \frac{\beta}{(t+1)^2}R^2 \right).$$

By smoothness we have:

$$\|\nabla F(\mathbf{y}_t)\|_2 \le \sqrt{2M(F(\mathbf{y}_t) - F(\mathbf{x}^*))}$$
$$= \sqrt{2M}c_2 \left( \sqrt{ \begin{matrix} \frac{1}{t^2} \sum_{i=0}^{t-1} \frac{c_{DP}Li^{3/2}\sqrt{\log(T/\gamma)} \|\nabla F(\mathbf{x}_i)\|_2}{\sqrt{B}\beta} \\ + \frac{c_{DP}LR \log(T/\gamma)}{\sqrt{B(t+1)}} + \frac{\beta b_{max}R}{t+1} + \frac{\beta R^2}{(t+1)^2} \end{matrix} } \right).$$

Then by smoothness and definition of $\mathbf{x}_t, \mathbf{y}_t$ we have:

$$\|\nabla F(\mathbf{x}_t)\|_2 \le M \|\mathbf{x}_t - \mathbf{y}_t\|_2 + \|\nabla F(\mathbf{y}_t)\|_2 \le 2MR/t + \|\nabla F(\mathbf{y}_t)\|_2 \le 2\sqrt{M\beta}R/t + \|\nabla F(\mathbf{y}_t)\|_2$$
$$\le \sqrt{2M}c_2 \left( \sqrt{ \begin{matrix} \frac{1}{t^2} \sum_{i=0}^{t-1} \frac{c_{DP}Li^{3/2}\sqrt{\log(T/\gamma)} \|\nabla F(\mathbf{x}_i)\|_2}{\sqrt{B}\beta} \\ + \frac{c_{DP}LR \log(T/\gamma)}{\sqrt{B(t+1)}} + \frac{\beta b_{max}R}{t+1} + \frac{\beta R^2}{(t+1)^2} \end{matrix} } + \frac{\sqrt{2\beta}R}{c_2 t} \right) \quad (20)$$

Isolating the sum inside the square root, we have by plugging in the inductive bound on $\|\nabla F(\mathbf{x}_i)\|_2$ and using $\sum_{i=0}^{t-1} i^k \le \int_0^t x^k dx = \frac{t^{k+1}}{k+1}$:

$$\frac{c_1 c_{DP} L \sqrt{M \log(T/\gamma)}}{t^2 \sqrt{B} \beta} \sum_{i=0}^{t-1} i^{3/2} \left( \sqrt{\frac{c_{DP} LR \log(T/\gamma)}{\sqrt{B(i+1)}}} + \sqrt{\frac{\beta b_{max} R}{i+1}} + \frac{2\sqrt{\beta}R}{i+1} \right)$$

$$\leq \frac{c_1 c_{DP} L \sqrt{M \log(T/\gamma)}}{\sqrt{B}\beta} \left( \frac{4t^{1/4}}{9} \sqrt{\frac{c_{DP} LR \log(T/\gamma)}{\sqrt{B}}} + \frac{\sqrt{\beta b_{max} R}}{2} + \frac{4\sqrt{\beta}R}{3\sqrt{t}} \right)$$

$$\leq \frac{4 c_1 \sqrt{M}}{3} \left( \frac{(c_{DP} L)^{3/2} \sqrt{R} \log(T/\gamma) t^{1/4}}{B^{3/4}\beta} + \frac{c_{DP} L \sqrt{\log(T/\gamma) b_{max} R}}{\sqrt{B}\beta} + \frac{c_{DP} LR \sqrt{\log(T/\gamma)}}{\sqrt{B\beta t}} \right) \tag{21}$$

We now use the assumptions in the lemma to simplify further:

$$\frac{4 c_1 \sqrt{M}}{3} \left( \frac{(c_{DP} L)^{3/2} \sqrt{R} \log(T/\gamma) t^{1/4}}{B^{3/4}\beta} + \frac{c_{DP} L \sqrt{\log(T/\gamma) b_{max} R}}{\sqrt{B}\beta} + \frac{c_{DP} LR \sqrt{\log(T/\gamma)}}{\sqrt{B\beta t}} \right)$$

$$\leq \frac{4 c_1 c_{DP} \sqrt{M} LR \log(T/\gamma)}{3} \left( \frac{t^{1/4}}{B^{3/4}\sqrt{c_{DP}}\beta} + \frac{p^{1/4}\sqrt{c_{DP} L c_{DP}}}{B\beta\sqrt{\varepsilon R}} + \frac{1}{\sqrt{B\beta t}} \right)$$

$$\leq \frac{4 c_1 \sqrt{c_{DP}} LR \log(T/\gamma)}{3} \left( \frac{t^{1/4}}{B^{3/4} c_{DP} \max\{1, \sqrt{p}/\varepsilon\sqrt{n}\}} \right.$$

$$\left. + \frac{p^{1/4}\sqrt{M}}{B \max\{\sqrt{\varepsilon}, p^{1/4}/n^{1/4}\}\sqrt{\beta c_{DP}}} + \frac{1}{\sqrt{c_{DP} B t}} \right)$$

$$\leq \frac{4 c_1 \sqrt{c_{DP}} LR \log(T/\gamma)}{3} \left( \frac{T}{n^{3/4} c_{DP}} + \frac{n^{1/4}}{B\sqrt{c_{DP}}} + \frac{1}{\sqrt{c_{DP} B t}} \right)$$

$$\leq \frac{4 c_1 \sqrt{c_{DP}} LR \log(T/\gamma)}{3} \left( \frac{2\sqrt{c_{DP}}}{n^{1/2}} + \frac{1}{\sqrt{c_{DP} B t}} \right)$$

$$\leq \frac{4 c_1 \sqrt{c_{DP}} LR \log(T/\gamma)}{3} \left( \frac{2\sqrt{c_{DP}}}{n^{1/2}} + \frac{1}{\sqrt{c_{DP} B t}} \right). \tag{22}$$

We can now plug Eq. (21) and Eq. (22) into Eq. (20), use subadditivity of the square root, and use the fact that for all sufficiently large $n$, $c_{DP} \geq c_M$:

$$\|\nabla F(\mathbf{x}_t)\|_2$$

$$\leq \sqrt{2M}c_2 \left( \sqrt{\begin{array}{l} \frac{8c_1 c_{DP} LR \log(T/\gamma)}{3} \cdot \frac{1}{n^{1/2}} + \frac{(4c_1/3c_{DP}+1)c_{DP} LR \log(T/\gamma)}{\sqrt{B(t+1)}} \\ + \frac{\beta b_{max} R}{t+1} + \frac{\beta R^2}{(t+1)^2} \end{array}} + \frac{\sqrt{2\beta}R}{c_2 t} \right)$$

$$\leq \sqrt{2M}c_2 \left( \sqrt{(8c_1\sqrt{c_\beta} + \frac{4c_1}{3c_{DP}} + 1)\frac{c_{DP} LR \log(T/\gamma)}{\sqrt{B(t+1)}} + \frac{\beta b_{max} R}{t+1} + \frac{\beta R^2}{(t+1)^2}} + \frac{\sqrt{2\beta}R}{c_2 t} \right)$$

$$\leq \sqrt{2M}c_2 \left( \sqrt{(8c_1 + \frac{4c_1}{3c_{DP}} + 1)\frac{c_{DP} LR \log(T/\gamma)}{\sqrt{B(t+1)}}} + \sqrt{\frac{\beta b_{max} R}{t+1}} + \sqrt{\frac{\beta R^2}{(t+1)^2}} + \frac{\sqrt{2\beta}R}{c_2 t} \right)$$

$$\leq \sqrt{2M}c_2 \left( \sqrt{(8c_1 + \frac{4c_1}{3c_{DP}} + 1)\frac{c_{DP} LR \log(T/\gamma)}{\sqrt{B(t+1)}}} + \sqrt{\frac{\beta b_{max} R}{t+1}} + \left(1 + \frac{\sqrt{2}}{c_2}\right)\frac{\sqrt{\beta}R}{t} \right)$$

$$\leq \sqrt{M} \left( \sqrt{2}c_2 \max\left\{ \sqrt{8c_1 + \frac{4c_1}{3c_{DP}} + 1}, 1 + \frac{\sqrt{2}}{c_2} \right\} \right)$$

$$\cdot \left( \sqrt{\frac{c_{DP} LR \log(T/\gamma)}{\sqrt{B(t+1)}}} + \sqrt{\frac{\beta b_{max} R}{t+1}} + \frac{\sqrt{\beta}R}{t} \right)$$

The inductive step is now proven if $c_1$ is such that:

$$\sqrt{2}c_2 \max\left\{ \sqrt{8c_1 + \frac{4c_1}{3c_{DP}} + 1}, 1 + \frac{\sqrt{2}}{c_2} \right\} \leq c_1$$

The left hand side has a sublinear dependence on $c_1$, and all other terms on the left hand side are constant except $1/c_{DP}$ which is decreasing in $n, 1/\delta$. So for a sufficiently large choice of $c_1$ and all sufficiently large $n, \delta \leq 1/n$, the inductive step holds. ∎

### 5.3. Bounding sensitivity

We now prove a sensitivity bound on each $\Delta_t$. The proof of sensitivity follows a similar structure to that of (Zhang et al., 2022b, Lemma 15). However, because the gradients $\nabla f_t(\mathbf{x}_t)$ are evaluated at a different point than that in Zhang et al. (2022b), the exact argument is different. In particular, we need the tight bound on gradient norms of Lemma 6 to ensure the sensitivity is constant while keeping $\beta$ roughly constant.

**Lemma 7 (Bounding $\ell_2$-sensitivity)** *In Algorithm 1, assume for all $t$ and $c_1$ as defined in Lemma 6,*

$$\|\nabla F(\mathbf{x}_t)\|_2 \leq c_1 \sqrt{M} \left( \sqrt{\frac{c_{DP} LR \log(T/\gamma)}{\sqrt{B(t+1)}}} + \sqrt{\frac{\beta b_{max} R}{t+1}} + \frac{2\sqrt{\beta}R}{t+1} \right).$$

*Then, using the parameters* (19), *for all sufficiently large $n$, $\gamma = \Omega(\delta)$, $\delta = 2^{-o(n^{1/3})}$, $p \leq \varepsilon^2 n^2$, under a probability $1 - 2\gamma$ event contained in the probability $1 - \gamma$ event of Theorem* 3, *letting $\Delta_t(d) := \eta_t \nabla f(\mathbf{x}_t, d) - \eta_{t-1} \nabla f(\mathbf{x}_{t-1}, d)$,*

$$\forall t \leq T - 1, d \in supp(\mathcal{D}) : \|\Delta_t(d)\|_2 \leq c_{DP} L.$$

**Proof** We have for any $d$:

$$\Delta_t(d) = \eta_t \left( \nabla f(\mathbf{x}_t, d) - \nabla f(\mathbf{x}_{t-1}, d) \right) + \left( \eta_t - \eta_{t-1} \right) \nabla f(\mathbf{x}_t, d)$$
$$\Rightarrow \|\Delta_t(d)\|_2 \leq \eta_t \cdot M \|\mathbf{x}_t - \mathbf{x}_{t-1}\|_2 + |\eta_t - \eta_{t-1}| \cdot L. \tag{23}$$

For $t \leq 1 + (c_{DP} - 1)/c_M$:

$$\eta_t \cdot M \|\mathbf{x}_t - \mathbf{x}_{t-1}\|_2 + |\eta_t - \eta_{t-1}| \cdot L$$
$$\leq \eta_t \cdot MR + L$$
$$\leq (\eta_t \cdot c_M + 1) + L$$
$$\leq c_{DP} L$$

We prove the bound for $t > 1 + (c_{DP} - 1)/c_M$ by induction. Assuming the bound holds for $0, 1, \ldots, t - 1$, we bound each of the terms in the R.H.S. of Equation (23) independently. Since $\eta_t = t + 1$, $\eta_t - \eta_{t-1} = 1$. In the following, we bound $\|\mathbf{x}_t - \mathbf{x}_{t-1}\|_2$. We use the fact that using 19,

for all sufficiently large $n$ we have $\beta \geq c_{DP}M$ to get:

$$\eta_{0:t} \cdot \mathbf{x}_t = \eta_{0:t-1} \cdot \mathbf{y}_t + \eta_t \cdot \mathbf{z}_t$$
$$\Leftrightarrow \eta_{0:t} \|\mathbf{x}_t - \mathbf{x}_{t-1}\|_2 = \|\eta_{0:t-1}(\mathbf{y}_t - \mathbf{x}_{t-1}) + \eta_t(\mathbf{z}_t - \mathbf{x}_{t-1})\|_2$$
$$\leq \eta_{0:t-1} \|\mathbf{y}_t - \mathbf{x}_{t-1}\|_2 + \eta_t \|\mathbf{z}_t - \mathbf{x}_{t-1}\|_2$$
$$\leq \eta_{0:t-1} \left\| \frac{\nabla_{t-1}}{\beta} + \frac{\mathbf{b}_{t-1}}{\eta_t} \right\|_2 + \eta_t R$$
$$\Rightarrow \|\mathbf{x}_t - \mathbf{x}_{t-1}\|_2 \leq \frac{\eta_{0:t-1}}{\eta_{0:t}} \cdot \frac{\|\nabla F(\mathbf{x}_{t-1})\|_2}{\beta}$$
$$+ \frac{\eta_{0:t-1}}{\eta_{0:t}} \cdot \frac{\|\nabla_{t-1} - \nabla F(\mathbf{x}_{t-1})\|_2}{\beta} + \frac{\eta_{0:t-1} \|\mathbf{b}_{t-1}\|_2}{\eta_t \cdot \eta_{0:t}} + \frac{\eta_t}{\eta_{0:t}} R$$
$$\overset{(*_1)}{\leq} \frac{c_1 \sqrt{M} \left( \sqrt{\frac{c_{DP}LR\log(T/\gamma)}{\sqrt{B(t+1)}}} + \sqrt{\frac{\beta b_{max}R}{t+1}} + \frac{2\sqrt{\beta}R}{t+1} \right)}{\beta}$$
$$+ \frac{\|\nabla_{t-1} - \nabla F(\mathbf{x}_{t-1})\|_2}{\beta} + \frac{1}{t+1} b_{max} + \frac{2}{t+2} R$$
$$\leq \frac{c_1 \sqrt{LR\log(T/\gamma)}}{\sqrt{\beta c_{DP}}(B(t+1))^{1/4}} + \frac{c_1 \sqrt{b_{max}R}}{\sqrt{c_{DP}(t+1)}} + \frac{2c_1 R}{(t+1)}$$
$$+ \frac{\|\nabla_{t-1} - \nabla F(\mathbf{x}_{t-1})\|_2}{\beta} + \frac{1}{t+1} b_{max} + \frac{2}{t+2} R$$
$$\Rightarrow \eta_t \cdot M \|\mathbf{x}_t - \mathbf{x}_{t-1}\|_2 \leq \frac{c_1 M(t+1)^{3/4}\sqrt{LR\log(T/\gamma)}}{\sqrt{\beta c_{DP}}B^{1/4}} + \frac{c_1 M\sqrt{b_{max}R(t+1)}}{\sqrt{c_{DP}}}$$
$$+ \eta_t \|\nabla_{t-1} - \nabla F(\mathbf{x}_{t-1})\|_2 + M b_{max} + (2c_1 + 2)MR. \qquad (24)$$

$(*_1)$ is by Lemma 6. We bound the first three terms individually. For the first term:

$$\frac{c_1 M(t+1)^{3/4}\sqrt{LR\log(T/\gamma)}}{\sqrt{\beta c_{DP}}B^{1/4}}$$
$$\leq \frac{c_1 MT^{3/4}\sqrt{LR\log(T/\gamma)}}{\sqrt{\beta c_{DP}}B^{1/4}}$$
$$= \frac{c_1 MT\sqrt{LR\log(T/\gamma)}}{\sqrt{\beta c_{DP}}n^{1/4}}$$
$$= \frac{c_1 MR\sqrt{\log(T/\gamma)}}{\sqrt{c_{DP}}\max\{1, \sqrt{p}/\varepsilon\sqrt{n}\}}$$
$$\leq c_1 MR\sqrt{\log(T/\gamma)}.$$

For the second term:

$$\frac{c_1 M \sqrt{b_{max} R(t+1)}}{\sqrt{c_{DP}}}$$

$$= c_1 M \sqrt{\frac{c_{DP} L R \sqrt{p}(t+1)}{\varepsilon B \beta}}$$

$$= c_1 M R \sqrt{\frac{\sqrt{p}(t+1)}{c_{DP} B \max\{\varepsilon, \sqrt{p}/\sqrt{n}\}}}$$

$$\leq c_1 M R \sqrt{\frac{\sqrt{n}(t+1)}{c_{DP} B}}$$

$$\leq c_1 M R \sqrt{\frac{\sqrt{n} T}{c_{DP} B}}$$

$$= c_1 M R \sqrt{\frac{\sqrt{n}(c_{DP} n^{1/4})}{c_{DP}(n^{3/4}/c_{DP})}}$$

$$= c_1 M R \sqrt{c_{DP}}.$$

For the third term:

$$\nabla_{t-1} - \nabla F(\mathbf{x}_{t-1}) = \frac{\sum_{i \leq t-1} \sum_{d \in \mathcal{B}_i} (\Delta_i(d) - \mathbb{E}_{d \sim \mathcal{D}}[\Delta_i(d)])}{Bt}$$

By the inductive hypothesis and triangle inequality, this is a vector martingale that is the sum of $Bt$ mean-zero terms each with norm at most $2c_{DP} L$. By a vector Azuma inequality (Hayes, 2005), w.p. $1 - \gamma$ (over all timesteps) we have:

$$\|\nabla_{t-1} - \nabla F(\mathbf{x}_{t-1})\|_2 \leq \frac{2 c_{DP} L \sqrt{3 \log(6T/\gamma)}}{\sqrt{Bt}}$$

Plugging these three bounds into Section 5.3 and the resulting bound into Equation (23), we have:

$$\|\Delta_t(d)\|_2 \leq c_1 M R \sqrt{\log(T/\gamma)} + c_1 M R \sqrt{c_{DP}}$$

$$+ \frac{2 c_{DP} L \sqrt{6T \log(6T/\gamma)}}{\sqrt{B}} + M b_{\max} + 2(c_1 + 1) M R + L.$$

$$\leq c_1 M R \sqrt{\log(T/\gamma)} + c_1 M R \sqrt{c_{DP}}$$

$$+ \frac{2 c_{DP}^2 L \sqrt{6 \log(6T/\gamma)}}{n^{1/4}} + (2c_1 + 3) M R + L.$$

$$\leq \left( c_1 c_M \sqrt{\log(T/\gamma)} + c_1 c_M \sqrt{c_{DP}} \right.$$

$$\left. + \frac{2 c_{DP}^2 \sqrt{6 \log(6T/\gamma)}}{n^{1/4}} + 2c_1 c_M + 3c_M + 1 \right) \cdot L. \tag{25}$$

(25) is at most $c_Q L$ and hence the lemma is proven by induction as long as:

$$c_1 c_M \sqrt{c_{DP} \log(T/\gamma)} + c_1 c_M \sqrt{c_{DP}} + \frac{2c_{DP}^2 \sqrt{6\log(6T/\gamma)}}{n^{1/4}} + 2c_1 c_M + 3c_M + 1 \le c_{DP}.$$

Since $c_1, c_M$ are constants, we have $\sqrt{c_{DP} \log(T/\gamma)} = o(c_{DP})$ since $c_{DP} = \omega(\sqrt{\log(T/\gamma)})$ under the condition $\gamma = \Omega(\delta)$, and we have $\frac{2c_{DP}^2 \sqrt{6\log(6T/\gamma)}}{n^{1/4}} = o(c_{DP})$ under the condition $\delta = 2^{-o(n^{1/3})}$. Hence the left-hand side is $o(c_{DP})$, so for all sufficiently large $n$ the inductive step holds. ∎

A crucial aspect of Lemma 7 is that conditioned on $b_{max}$, the sensitivity is independent of $t$, even if the norm of the quantity that is being approximated (i.e., $\eta_t \nabla f(\mathbf{x}_t; d)$) scales with $t$.

**Deciding the noise scale for $(\varepsilon, \delta)$-DP:** We now show how to sample $\mathbf{b}_t$ such that Algorithm 1 satisfies $(\varepsilon, \delta)$-DP while achieving strong utility guarantees. Each $\nabla_t$ is a (rescaling of a) prefix sum of the $\Delta_t$ values. So, choosing a noise mechanism that does not add too much noise to the gradients in Algorithm 1 is equivalent to privatizing the list of prefix sums $\{\Delta_1, \Delta_1 + \Delta_2, \dots, \Delta_1 + \dots + \Delta_T\}$, where by Claim 7, we know the sensitivity of each $\Delta_t$ is constant with respect to $t$. This is a well studied problem known as private continual counting, and a standard solution to this problem is the binary tree mechanism (Dwork et al., 2010).

**Lemma 8** *Let $\mathbf{g}_i = \sum_{j \le i} \Delta_i$ for $i \in [T]$ and suppose the $\Delta_i$ are computed such that between any two adjacent datasets, only one $\Delta_i$ changes, and it changes by $\ell_2$-norm at most $C$. Then for any $T \ge 2$ there is a $\mu$-GDP algorithm (i.e., an algorithm which satisfies any DP guarantee satisfied by the Gaussian mechanism with sensitivity $\mu$ and variance 1, including $(\mu\sqrt{2\ln(1.25/\delta)}, \delta)$-DP for all $\delta > 0$ (Dong et al., 2019)) that gives an unbiased estimate of each $\mathbf{g}_i$ such that with probability $1 - \delta$ all $\mathbf{g}_i$'s estimates have $\ell_2$-error at most $\frac{4C \log^{3/2} T \sqrt{p \ln(2T/\delta)}}{\mu}$.*

**Proof** For completeness, we define the binary tree mechanism and state its guarantees here. For $k = 0, 1, 2, \dots, \lceil \log T \rceil$, $j = 1, 3, 5, \dots T/2^k - 1$, the binary tree mechanism computes $s_{j,k} = \sum_{i=(j-1)\cdot 2^k+1}^{j \cdot 2^k} \Delta_i + \xi_{j,k}, \xi_{j,k} \overset{i.i.d.}{\sim} N(0, \sigma^2 \mathbb{I})$. An unbiased estimate of any prefix sum can be formed by adding at most $\lceil \log T \rceil$ of the $s_{j,k}$. For example, the sum of $\Delta_1$ to $\Delta_7$ has an unbiased estimator using at most $\lceil \log 7 \rceil = 3$ of the $s_{j,k}$, $s_{1,2} + s_{3,1} + s_{7,0}$.

Each $\Delta_i$ contributes to at most one $s_{j,k}$ for each value of $k$, i.e. at most a total of $1 + \lceil \log T \rceil$ values of $s_{j,k}$. So, if $\|\Delta_i\|_2 \le C$ for all $i$, then with $\sigma = C\sqrt{1 + \lceil \log T \rceil}/\mu$ we can release all $s_{j,k}$ with $\mu$-GDP. Furthermore, by a standard Gaussian tail bound with probability $1 - \delta/2$ we have that $\|\xi_i\|_2 \le \sigma \sqrt{2p \ln(2T/\delta)}$ for all $i$. Since each $\mathbf{g}_i$ has an unbiased estimator formed by adding at most $\lceil \log T \rceil$ of the $s_{j,k}$, the maximum additive error is then at most

$$\lceil \log T \rceil \max_i \|\xi_i\|_2 \le 2C \log T \sqrt{2p(1 + \lceil \log T \rceil) \ln(2T/\delta)}/\mu \le 4C \log^{3/2} T \sqrt{p \ln(2T/\delta)}/\mu.$$

∎

**Corollary 9** *Suppose we compute $\mathbf{b}_i$ in Algorithm 1 using the binary tree mechanism (Lemma 8) with $\sigma = \frac{c_{DP}L\sqrt{\log(T)\ln(2.5/\delta)}}{\varepsilon B\beta}$ to compute prefix sums of the $\Delta_i$. Then for the parameter choices in Claim 7, Algorithm 1 satisfies $(\varepsilon,\delta)$-DP and we have w.p. $1-\delta$:*

$$\max_t \|\mathbf{b}_t\|_2 \leq c_{DP}^2 L\sqrt{p}/\varepsilon B\beta.$$

**Proof** If we run Algorithm 1 using the binary tree mechanism to generate the $\mathbf{b}_i$ but instead clip $\eta_t \nabla f(\mathbf{x}_t, d) - \eta_{t-1}\nabla f(\mathbf{x}_{t-1}, d)$ for each $t$ and $d$ in $\mathcal{B}_t$ to norm at most $c_{DP}L$ if its norm exceeds $c_{DP}L$, then by Lemma 8 Algorithm 1 satisfies $\varepsilon/\sqrt{2\ln(2.5/\delta)}$-GDP which implies it satisfies $(\varepsilon,\delta/2)$-DP. Algorithm 1 without clipping is within total variation distance $\delta/2$ of the version with clipping by Lemma 6 and Lemma 7, i.e. Algorithm 1 satisfies $(\varepsilon,\delta)$-DP. By Lemma 8, plugging in $C = c_{DP}L$ and $\mu = \varepsilon/\sqrt{2\ln(2.5/\delta)}$ we have:

$$
\begin{aligned}
\max_t \|\mathbf{b}_t\|_2 \leq 2\sigma\sqrt{2p\ln(2T/\delta)} &= \frac{2c_{DP}L\sqrt{2p\log(T)\ln(2.5/\delta)\ln(2T/\delta)}}{\varepsilon B\beta} \\
&\leq \frac{2c_{DP}L\sqrt{2p\log(n)\ln(2.5/\delta)\ln(2n/\delta)}}{\varepsilon B\beta} \\
&\leq \frac{2c_{DP}L\sqrt{2p\log(1/\delta)\ln(2.5/\delta)\ln(2n/\delta)}}{\varepsilon B\beta} \\
&\leq \frac{4c_{DP}L\ln(2.5/\delta)\sqrt{2p\ln(2n/\delta)}}{\varepsilon B\beta} \\
&\leq \frac{c_{DP}^2 L\sqrt{p}}{\varepsilon B\beta}.
\end{aligned}
$$

The bound on $\|\mathbf{b}_t\|_2$ also follows by Lemma 8. ■

Finally, we plug the error guarantees of the binary tree mechanism into Theorem 3 to get our main result:

**Theorem 10** *For $2^{-o(n^{1/6})} \leq \delta \leq 1/n$, $p \leq \varepsilon^2 n^2$, Algorithm 1 with parameters given by Eq. (19) and using the binary tree mechanism to sample $\mathbf{b}_i$ is $(\varepsilon,\delta)$-DP and has the following excess population risk guarantee:*

$$\mathbb{E}\left[F(\mathbf{y}_T)\right] - F(\mathbf{x}^*) = O\left(LR \cdot polylog(n, 1/\delta) \cdot \left(\frac{1}{\sqrt{n}} + \frac{\sqrt{p}}{\varepsilon n}\right)\right).$$

Before we prove the theorem, we note that both the lower bound on $\delta$ and upper bound on $p$ are vacuous: If $p \geq \varepsilon^2 n^2$, then $LR\sqrt{p}/\varepsilon n \geq LR$, and $LR$ is the excess risk achieved by picking any point in $\mathcal{C}$, i.e. the excess risk guarantee in the Theorem is trivially achieved. Similarly, if $\delta = 2^{-\Omega(n^{1/6})}$, then $\sqrt{n} \leq polylog(1/\delta)$ and again the excess risk guarantee in the Theorem is trivially achieved.

**Proof**

In Lemma 7 we chose $\gamma = \delta/4 \leq 1/4n$, in which case the high probability events of Theorem 3, Lemma 6 and Lemma 7 happen w.p. at least $1 - 1/2n$. We give a bound conditioned on

these events, which can differ from the unconditional final bound by at most $O(LR/n)$, which is a lower-order term.

From Theorem 3, using $Q = 2c_{DP}L$, $\|\mathbf{b}_t\|_2 \leq b_{max}$, and the bound from Lemma 6 on $\|\nabla F(\mathbf{x}_t)\|_2$, simplifying by suppressing terms logarithmic in $n, 1/\delta$ we have conditioned on the event of Lemma 7:

$$
F(\mathbf{y}_T) - F(\mathbf{x}^*) =
$$

$$
\widetilde{O}_{n,1/\delta} \left( \frac{1}{T^2} \sum_{t=1}^{T} t^2 \left( \frac{L \left( \sqrt{M} \left( \sqrt{\frac{LR}{\sqrt{B}t}} + \sqrt{\frac{\beta b_{max}R}{t}} + \frac{\sqrt{\beta}R}{t} \right) + \frac{L}{\sqrt{B}t} \right)}{\sqrt{B}t\beta} \right. \right.
$$

$$
\left. \left. + \frac{LR}{\sqrt{B}t^{3/2}} + \frac{\beta b_{max}R}{t^2} \right) + \frac{\beta}{T^2}R^2 \right)
$$

In (21) and (22) we bounded the first term inside the sum, we can reuse the bound to get:

$$
F(\mathbf{y}_T) - F(\mathbf{x}^*)
$$

$$
= \widetilde{O}_{n,1/\delta} \left( \frac{LR}{\sqrt{n}} + \frac{1}{T^2} \sum_{t=1}^{T} t^2 \left( \frac{LR}{\sqrt{B}t^{1.5}} + \frac{\beta b_{max}R}{t^2} \right) + \frac{\beta}{T^2}R^2 \right)
$$

$$
= \widetilde{O}_{n,1/\delta} \left( \frac{LR}{\sqrt{n}} + \frac{LR}{\sqrt{BT}} + \frac{\beta b_{max}R}{T} + \frac{\beta}{T^2}R^2 \right)
$$

$$
= \widetilde{O}_{n,1/\delta} \left( \frac{LR}{\sqrt{n}} + \frac{\beta b_{max}R}{T} + \frac{\beta}{T^2}R^2 \right).
$$

Finally, we plug in the parameter values from (19) and use the fact that $\beta b_{max} \leq c_{DP}^2 L\sqrt{p}/\varepsilon B$ to get:

$$
F(\mathbf{y}_T) - F(\mathbf{x}^*) = \widetilde{O}_{n,1/\delta} \left( LR \left( \sqrt{n} + \frac{\sqrt{p}}{\varepsilon n} \right) \right).
$$

$\blacksquare$

## 5.4. Removing the Assumption $\mathbf{x}^\dagger \in \mathcal{C}$

In this section we show how to modify our analysis to the setting where $\mathbf{x}^\dagger$ may lie outside the constraint set, i.e. the gradient at $\mathbf{x}^*$ is not necessarily 0. In this setting the gradient norm is no longer guaranteed to be decreasing in $t$, which causes our sensitivity to increase and also worsens our final utility bound. To undo both of these effects, we increase $\beta$ to be $\widetilde{\Theta}(MT)$ instead of $\widetilde{\Theta}(M)$, and consequently need to raise $T$ to be $\sqrt{n}$ instead of $\widetilde{\Theta}(n^{1/4})$ to ensure the optimization error $\beta R^2/T^2$ is $O(1/\sqrt{n})$.

**Lemma 11 (Bounding $\ell_2$-sensitivity)** *In Algorithm 1, suppose for all $t$, $\|\mathbf{b}_t\|_2 \leq b_{\max}$. Then for $\eta_t = t + 1$, $\beta \geq 2MT$ for any $d \in \mathcal{D}$, $t \in \{0, 1, \ldots, T-1\}$*

$$\|\eta_t \nabla f(\mathbf{x}_t, d) - \eta_{t-1} \nabla f(\mathbf{x}_{t-1}, d)\|_2 \leq 2(Mb_{\max} + 2MR + L) \leq 2Mb_{max} + (2c_M + 1)L.$$

**Proof** For convenience let $\Delta_t(d) = \eta_t \nabla f(\mathbf{x}_t, d) - \eta_{t-1} \nabla f(\mathbf{x}_{t-1}, d)$. The bound holds by Lipschitzness for $t = 0$. We prove the bound for $t > 0$ by induction. We have:

$$\Delta_t(d) = \eta_t \left(\nabla f(\mathbf{x}_t, d) - \nabla f(\mathbf{x}_{t-1}, d)\right) + (\eta_t - \eta_{t-1})\nabla f(\mathbf{x}_t, d)$$
$$\Rightarrow \|\Delta_t(d)\|_2 \leq \eta_t \cdot M \|\mathbf{x}_t - \mathbf{x}_{t-1}\|_2 + |\eta_t - \eta_{t-1}| \cdot L. \tag{26}$$

We now bound each of the terms in the R.H.S. of Equation (26) independently. Since $\eta_t = t + 1$, $\eta_t - \eta_{t-1} = 1$. In the following, we bound $\|\mathbf{x}_t - \mathbf{x}_{t-1}\|_2$. We have,

$$\eta_{0:t} \cdot \mathbf{x}_t = \eta_{0:t-1} \cdot \mathbf{y}_t + \eta_t \cdot \mathbf{z}_t$$
$$\Leftrightarrow \eta_{0:t} \|\mathbf{x}_t - \mathbf{x}_{t-1}\|_2 = \|\eta_{0:t-1}(\mathbf{y}_t - \mathbf{x}_{t-1}) + \eta_t(\mathbf{z}_t - \mathbf{x}_{t-1})\|_2$$
$$\leq \eta_{0:t-1} \|\mathbf{y}_t - \mathbf{x}_{t-1}\|_2 + \eta_t \|\mathbf{z}_t - \mathbf{x}_{t-1}\|_2$$
$$\leq \eta_{0:t-1} \left\|\frac{\nabla_{t-1}}{\beta} + \frac{\mathbf{b}_{t-1}}{\eta_t}\right\|_2 + \eta_t R$$
$$\Rightarrow \|\mathbf{x}_t - \mathbf{x}_{t-1}\|_2 \leq \frac{\eta_{0:t-1}}{\eta_{0:t}} \cdot \frac{\|\nabla_{t-1}\|_2}{\beta} + \frac{\eta_{0:t-1} \|\mathbf{b}_{t-1}\|_2}{\eta_t \cdot \eta_{0:t}} + \frac{\eta_t}{\eta_{0:t}} R$$
$$\leq \frac{\sum_{i \leq t-1} \sum_{d \in \mathcal{B}_i} \|\Delta_i(d)\|_2}{\beta(t+1)B} + \frac{1}{t+1} b_{\max} + \frac{2}{t+2} R$$
$$\leq \frac{\max_{i \leq t-1, d \in \mathcal{D}} \|\Delta_i(d)\|_2}{\beta} + \frac{1}{t+1} b_{\max} + \frac{2}{t+2} R$$
$$\Rightarrow \eta_t \cdot M \|\mathbf{x}_t - \mathbf{x}_{t-1}\|_2 \leq \frac{\eta_t M \cdot \max_{i \leq t-1, d \in \mathcal{D}} \|\Delta_i(d)\|_2}{\beta} + Mb_{\max} + 2MR$$
$$\leq \frac{\max_{i \leq t-1, d \in \mathcal{D}} \|\Delta_i(d)\|_2}{2} + Mb_{\max} + 2MR. \tag{27}$$

Plugging Equation (27) and the fact that $\eta_t - \eta_{t-1} = 1$ into Equation (26), we have the following:

$$\|\Delta_t(d)\|_2 \leq \frac{\max_{i \leq t-1, d \in \mathcal{D}} \|\Delta_i(d)\|_2}{2} + Mb_{max} + 2MR + L. \tag{28}$$

By induction, this completes the proof of Lemma 11. $\blacksquare$

By using the high-probability upper bound on $\mathbf{b}_t$ combined with Lemma 11, we can show that $Mb_{max}$ is a lower-order term in Lemma 11:

**Corollary 12** *Suppose we compute $\mathbf{b}_i$ in Algorithm 1 using the binary tree mechanism (Lemma 8) with $C = \frac{(8c_M + 4)L}{B\beta}$. Then for $p \leq \frac{B^2 \beta^2 \varepsilon^2}{32M^2 \log^3(T) \ln(2T/\delta) \ln(2.5/\delta)}$ with probability $1 - \delta/2$, $\|\Delta_t(d)\|_2 \leq (8c_M + 4)L$ for all $d \in supp(\mathcal{D}), t$.*

**Proof** The corollary follows from Lemma 11 if $b_{max} \leq (2c_M + 1)L/M$. Plugging $C$ into the bound on $\max_t \|\mathbf{b}_t\|_2$ from Lemma 8, we have:

$$b_{max} \leq \frac{(8c_M + 4)L \log^{3/2} T \sqrt{p \ln(2T/\delta)}}{B\beta\mu}.$$

Hence it suffices if:

$$\frac{(8c_M + 4)L \log^{3/2} T \sqrt{p \ln(2T/\delta)}}{B\beta\mu} \leq (2c_M + 1)L/M.$$

Rearranging, we get that it suffices if

$$p \leq \left(\frac{(2c_M + 1)B\beta\mu}{M(8c_M + 4) \log^{3/2} T \sqrt{\ln(2T/\delta)}}\right)^2 = \frac{B^2\beta^2\mu^2}{16M^2 \log^3(T) \ln(2T/\delta)}.$$

∎

Then by a similar argument to Corollary 9, we have:

**Corollary 13** *Suppose we compute $\mathbf{b}_i$ in Algorithm 1 using the binary tree mechanism (Lemma 8) with $\sigma = \frac{(16\sqrt{2}c_M + 8\sqrt{2})L\sqrt{\log(T) \ln(2.5/\delta)}}{\varepsilon B\beta}$ on $\Delta_i$. Then for $p \leq \frac{B^2\beta^2\mu^2}{16M^2 \log^3(T) \ln(2T/\delta)}$, Algorithm 1 satisfies $(\varepsilon, \delta)$-DP.*

While we require a bound on the dimension to ensure the norm of the noise is bounded, this bound will end up being nearly vacuous: Our choice of $\beta$ is proportional to $n^{3/2}B^2$, i.e. the bound we get is roughly $p \leq n^3\varepsilon^2/B^2$. We will choose $B \leq \sqrt{n}$ so it suffices if $p \leq \varepsilon^2 n^2$; this is the range of dimensions where DP-SCO is non-trivial, as for $p > \varepsilon^2 n^2$ any point in $\mathcal{C}$ achieves the (asymptotic) optimal loss bound.

We now give our main result without the assumption $\mathbf{x}^\dagger \in \mathcal{C}$:

**Theorem 14** *Algorithm 1 with batch size $B \leq \sqrt{n}$, $T = n/B$, $\eta_t = t + 1$, $\beta = \frac{(16c_M + 8)Ln^{3/2}}{RB^2}$, $\sigma = \frac{((16\sqrt{2}c_M + 8\sqrt{2})L)\sqrt{\log(T) \ln(2.5/\delta)}}{\varepsilon B\beta}$ is $(\varepsilon, \delta)$-DP and has the following excess population loss guarantee:*

$$\mathbb{E}\left[F(\mathbf{y}_T)\right] - F(\mathbf{x}^*) =$$

$$O\left(\frac{MR^2B^2}{n^2} + \frac{(L + MR)R}{\sqrt{n}} + \frac{(L + \frac{MR}{\sqrt{B}})R\sqrt{p}}{\varepsilon n} \cdot polylog(n/B, 1/\delta)\right).$$

*In particular, if we choose $B = \min\{\sqrt{n}, Mn^{3/2}\varepsilon/(\sqrt{p} \cdot polylog(n/B, 1/\delta))\}$ then Algorithm 1 requires*

$$\max\{\sqrt{n}, \sqrt{p} \cdot polylog(n/B, 1/\delta)/M\varepsilon\sqrt{n}\}$$

*batch gradients and:*

$$\mathbb{E}\left[F(\mathbf{y}_T)\right] - F(\mathbf{x}^*) = O\left(LR \cdot \left(\frac{1}{\sqrt{n}} + \frac{\sqrt{p}}{\varepsilon n} \cdot polylog(n/B, 1/\delta)\right)\right).$$

**Proof**

We assume $p = O\left(\frac{\varepsilon^2 n^2}{\text{polylog}(n,1/\delta)}\right)$; if $p = \omega\left(\frac{\varepsilon^2 n^2}{\text{polylog}(n,1/\delta)}\right)$, then any point in $\mathcal{C}$ achieves the desired excess risk. Under this assumption, the upper bound on $B$ gives

$$\beta \geq 16Mn^{3/2}/B^2 \geq 16Mn/B \geq 2MT,$$

satisfying the assumption in Claim 11, and also gives that $d$ satisfies the assumption in Corollary 13. Let $Q = O(L)$ in Theorem 3 by Corollary 12 and $b_{max} = \widetilde{O}_{n,1/\delta}\left(\frac{L\sqrt{p}}{\varepsilon B \beta}\right)$ by Lemma 8. We plug the tail bound of Lemma 8 into the high-probability error bound of Theorem 3, and use a failure probability of $1/T^2$ in both Lemma 8 and Theorem 3 (the increase in the expected error due to this low-probability event is $O(LR/T^2)$, which is a lower order term in our error bound). Then we have:

$$F(\mathbf{y}_T) - F(\mathbf{x}^*)$$

$$= \widetilde{O}_{n,1/\delta}\left(\frac{1}{T^2}\sum_{t=1}^{T} t^2 \left(\frac{L^2}{\sqrt{Bt}\beta} + \frac{Lb_{max}}{\sqrt{B}t^{1.5}} + \frac{LR}{\sqrt{B}t^{1.5}} + \frac{\beta b_{max}R}{t^2} + \frac{\beta b_{max}^2}{t^2}\right) + \frac{\beta R^2}{T^2}\right)$$

$$= \widetilde{O}_{n,1/\delta}\left(\frac{L^2\sqrt{T}}{\sqrt{B}\beta} + \frac{Lb_{max}}{\sqrt{BT}} + \frac{LR}{\sqrt{BT}} + \frac{\beta b_{max}R}{T} + \frac{\beta b_{max}^2}{T} + \frac{\beta R^2}{T^2}\right)$$

$$= \widetilde{O}_{n,1/\delta}\left(\frac{LRB^{1.5}\sqrt{T}}{n^{3/2}} + \frac{L^2\sqrt{p}}{\varepsilon\beta B^{1.5}\sqrt{T}} + \frac{LR}{\sqrt{BT}} + \frac{LR\sqrt{p}}{\varepsilon BT} + \frac{L^2 p}{\varepsilon^2 B^2 \beta T} + \frac{n^{3/2}LR}{B^2T^2}\right).$$

$$= \widetilde{O}_{n,1/\delta}\left(\frac{LRB}{n} + \frac{LR\sqrt{Bp}}{\varepsilon n^{3/2}\sqrt{T}} + \frac{LR}{\sqrt{n}} + \frac{LR\sqrt{p}}{\varepsilon n} + \frac{LRp}{\varepsilon^2 n^{3/2}T} + \frac{LR}{\sqrt{n}}\right).$$

$$\stackrel{(*_1)}{=} \widetilde{O}_{n,1/\delta}\left(\frac{LR}{\sqrt{n}} + \frac{LR\sqrt{p}}{\varepsilon n} + \frac{LRp}{\varepsilon^2 n^2}\right).$$

$$\stackrel{(*_2)}{=} \widetilde{O}_{n,1/\delta}\left(\frac{LR}{\sqrt{n}} + \frac{LR\sqrt{p}}{\varepsilon n}\right).$$

In $(*_1)$, we use the fact that $B \leq \sqrt{n}$ and $T \geq \sqrt{n}$, and in $(*_2)$ we use the assumption $p = O\left(\frac{\varepsilon^2 n^2}{\text{polylog}(n,1/\delta)}\right)$, which implies $p/\varepsilon^2 n^2 \leq \sqrt{p}/\varepsilon n$. $\blacksquare$

### 5.5. Analysis of Stochastic Accelerated Gradient Descent using Independent Gradients

**Theorem 15** *If we use $\nabla_t = \frac{1}{B}\sum_{d\in\mathcal{B}_t}\nabla f(\mathbf{x}_t, d)$ instead in* `Accelerated-DP-SRGD`*, if $\eta_t^2 \leq 4\eta_{0:t}$ and $\beta \geq M$ then we have:*

$$\mathbb{E}\left[F(\mathbf{y}_T)\right] - F(\mathbf{x}^*) \leq \frac{1}{\eta_{0:T}}\sum_{i=0}^{T-1}\left[\eta_{0:t}\left(\frac{8L^2}{\sqrt{B}\beta} + \frac{5L\|\mathbf{b}_t\|_2}{\eta_t} + 3\beta\|\mathbf{b}_t\|_2 \cdot R + \frac{\beta\|\mathbf{b}_t\|_2^2}{\eta_t^2}\right)\right] + \frac{\beta}{\eta_{0:T}}R^2.$$

**Proof**

Each $\mathbf{q}_t$ is now the average of $B$ random vectors which have norm at most $2L$. So, we have $\mathbb{E}_{\mathbf{q}_t}\left[\|\mathbf{q}_t\|_2^2 \,|\mathbf{x}_t\right] \leq \frac{4L^2}{B}$. By Jensen's inequality, this gives $\mathbb{E}_{\mathbf{q}_t}\left[\|\mathbf{q}_t\|_2 \,|\mathbf{x}_t\right] \leq \frac{2L}{\sqrt{B}}$. Now, using the fact that $\mathbf{q}_t$ is mean zero conditioned on $\mathbf{x}_t$, we have (using $\mathbf{y}'_{t+1}$ as defined in the proof of Theorem 3):

$$\mathbb{E}_{\mathbf{q}_t}\left[-\left\langle \mathbf{q}_t, \mathbf{y}'_{t+1} - \mathbf{x}_t \right\rangle |\mathbf{x}_t\right] = \frac{1}{\beta}\mathbb{E}_{\mathbf{q}_t}\left[\|\mathbf{q}_t\|_2^2 \,|\mathbf{x}_t\right] \leq \frac{4L^2}{B\beta}. \tag{29}$$

Combining Equation (11), Equation (29), and the fact that now $\|\nabla_t\|_2 \leq L$ unconditionally, we have the following:

$$-\eta_{0:t}\mathbb{E}_{\mathbf{q}_t}\left[\left\langle \mathbf{q}_t, \mathbf{y}_{t+1} - \mathbf{x}_t \right\rangle |\mathbf{x}_t\right] \leq \eta_{0:t}\left(\frac{8L^2}{\sqrt{B}\beta} + \frac{4L\|\mathbf{b}_t\|_2}{\sqrt{B}\eta_t}\right). \tag{30}$$

Using the fact that $\mathbf{q}_t$ is mean-zero and $\mathbb{E}_{\mathbf{q}_t}\left[\|\mathbf{q}_t\|_2 \,|\mathbf{x}_t\right] \leq \frac{2L}{\sqrt{B}}$ we then have the following bound on the potential change by plugging (30) into (10):

$$\mathbb{E}_{\mathbf{q}_t}\left[\Delta\Phi_t |\mathbf{x}_t\right] \leq \eta_{0:t}\left(\frac{8L^2}{\sqrt{B}\beta} + \frac{5L\|\mathbf{b}_t\|_2}{\eta_t} + 3\beta\|\mathbf{b}_t\|_2 \cdot R + \frac{\beta\|\mathbf{b}_t\|_2^2}{\eta_t^2}\right) \tag{31}$$

We now proceed similarly to Theorem 3. ■

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
