# OpenReview forum: "Optimal Rates for O(1)-Smooth DP-SCO with a Single Epoch and Large Batches"
_algorithmiclearningtheory.org/ALT/2025/Conference — ALT 2025_

### Official Review · Reviewer_RdzJ · 2024-10-30
**Review for Submission 58**

**Rating:** 7
**Confidence:** 4

**Review:**

This paper studies the problem of Differentially Private Stochastic Convex Optimization. This is a well-studied problem for which optimal excess population risk rates have been known for 5 years. Its main contribution is in reducing the number of adaptive steps of the algorithm --called batch gradient complexity in the submission-- (and relatedly increasing batch sizes) for a single pass algorithm. Moreover, their privacy analysis does not directly rely on convexity, which makes it suitable for its use in nonconvex learning problems.

The key insights of this algorithm are in the use of the Nesterov acceleration in conjunction with (stochastic) variance reduction. These two ideas have not been systematically studied in this context (at least, not together) which already makes the analyses of the paper interesting. The paper also provides an improved  rate for the batch gradient complexity (of order $n^{1/4}$) under the additional assumption that the optimal solution is interior, which is based on a novel sensitivity analysis tied to the reduction of the norm of the gradient in this case.

I believe this paper contributes significantly on the theoretical and conceptual side. While the rates do not appear to be novel, there are good reasons to be interested in reducing the number of adaptive steps (e.g., it potentially reduces lags in distributed environments, and mitigates the impact of composition for privacy guarantees). I also believe that some of the techniques introduced in this work may be of independent interest. On the downside, I believe some claims in the paper are not very clear, and I would like to see a final version that is more careful in this respect.

Minor comments:
1. The submission claims to obtain the optimal rates for DP-SCO, but the analysis introduces polylog(n) factors which are not present in the lower bounds. Moreover, the established rates hold with high probability, so these factors may be unavoidable. Can the authors clarify this?
2. Still on this issue of polylogs, it is claimed in page 2 that "even without assuming $x^+\in{\cal C}$ our work improves upon all prior works (at least by $polylog(n)$"). Is this claim in terms of the gradient complexity of excess risk. For the latter, it is very unclear to me that this is the case, for the reasons stated in the previous point.
3. In the same vein, I was expecting a high probability bound in the final result (theorem 10), but the excess risk is only stated in expectation.
4. In the context of DP-SCO, algorithms based on the stochastic Frank-Wolfe method use the stochastic recursive gradient estimators (Asi et al.2021a, Bassily et al. 2021). While the current submission uses this idea in a different context (stochastic acceleration), I believe this is a relevant comparison which is currently missing.
5. The submission mentions that "To the best of our knowledge, a direct potential-based analysis of stochastic accelerated gradient descent has not appeared in the literature before besider (Taylor and Bach,2019)." On the one hand, such analyses exist (e.g., Cohen, Diakonikolas, Orecchia http://proceedings.mlr.press/v80/cohen18a/cohen18a.pdf). On the other hand, the analysis of Taylor and Bach is specific to randomization by subsampling, which is orthogonal to the stochastic i.i.d. analysis carried out in the submission.

**Paper Award:**

No

---

> ### Author Response · Authors · 2024-11-22
> **Response to reviewer RdzJ**
>
> Thanks to the reviewer for their suggestions on improving the paper. Below we respond to the comments in the review:
>
> **The submission claims to obtain the optimal rates for DP-SCO, but the analysis introduces polylog(n) factors which are not present in the lower bounds. Moreover, the established rates hold with high probability, so these factors may be unavoidable. Can the authors clarify this?**
> * Our approach inherently incurs polylog factors, even if we are not aiming for high probability utility bounds (e.g., because we need a tail bound on certain quantities in order to bound the sensitivity, and the corresponding low probability event is absorbed into the privacy guarantee). We will make this clearer in the introduction and discussion of the corresponding proofs.
> *We try to be explicit about the fact that we are optimal only up to polylog factors in the introduction, we will take another pass and correct any instances where this may be unclear. We are also happy to change the title to “Near-Optimal” if the reviewer feels “Optimal” is too confusing or misleading.
>
> **Still on this issue of polylogs, it is claimed in page 2 that "even without assuming $x^\dagger \in C$ our work improves upon all prior works (at least by $polylog(n)$ "). Is this claim in terms of the gradient complexity of excess risk. For the latter, it is very unclear to me that this is the case, for the reasons stated in the previous point.**
> * This improvement is in batch gradient complexity; we will restate this explicitly in the paragraph.
>
> **In the same vein, I was expecting a high probability bound in the final result (theorem 10), but the excess risk is only stated in expectation.**
> * We do prove a high-probability bound, but we ultimately convert it into an expectation bound for an easier comparison to past DP-SCO works such as Bassily et al. 2014 and Feldman et al. 2020 which use expectation bounds as well. We will make it explicit that we achieve both high-probability bounds and expectation bounds for the excess risk.
>
> **In the context of DP-SCO, algorithms based on the stochastic Frank-Wolfe method use the stochastic recursive gradient estimators (Asi et al.2021a, Bassily et al. 2021). While the current submission uses this idea in a different context (stochastic acceleration), I believe this is a relevant comparison which is currently missing.**
> * Thanks for the pointers. We will add a discussion of these results using private variance-reduction estimators to our discussion in the intro on SRGs.
>
> **The submission mentions that "To the best of our knowledge, a direct potential-based analysis of stochastic accelerated gradient descent has not appeared in the literature before besides (Taylor and Bach,2019)." On the one hand, such analyses exist (e.g., Cohen, Diakonikolas, Orecchia http://proceedings.mlr.press/v80/cohen18a/cohen18a.pdf). On the other hand, the analysis of Taylor and Bach is specific to randomization by subsampling, which is orthogonal to the stochastic i.i.d. analysis carried out in the submission.**
> * Thank you for the pointer to the Cohen et al. work. We agree that they too essentially provide a potential argument, and that ours is not the only work besides Taylor and Bach, and will revise our claim accordingly.

---

> > ### Comment · Reviewer_RdzJ · 2024-11-26
> >
> > Thank you for the clarifications. I agree on changing the title to Near-Optimal.
> >
> > I am maintaining my score.

---

### Official Review · Reviewer_Ggyu · 2024-11-08
**Comments on Paper 58**

**Rating:** 6
**Confidence:** 3

**Review:**

#Strength
1) This paper first proposes a new algorithm for DP stochastic convex optimization, namely Accelerated-DP-SRGD, based on a careful adaptation of Nesterov’s accelerated stochastic gradient descent.
2) The proposed algorithm can achieve optimal DP-SCO errors for $O(1)$-smooth convex losses with $\sqrt{n}$ batch gradient steps, which is better than the prior works in this term.
3) This paper analyzes the cases where the unconstrained population risk minimizer $\mathbf{x}^{\dagger}$ is and is not within the constraint set $\mathcal{C}$. When it is within $\mathcal{C}$, the batch gradient steps can be further reduced to $n^{1/4}$.
#Weakness
1) The structure of the paper is not well-organized. The introduction section is too long, i.e., over 7 pages.
2) The comparison with related work is limited, which fails to clearly explain the challenges in reducing batch gradient complexity.
#Suggestions
1) It would be better if you introduce the definitions of $L$-Lipschitz and $M$-smooth functions before presenting the problem definition in page 2.
2) In line 6 of the second paragraph on page 3, "number of" is written twice.
3) There's quite a bit of blank space on page 22.
4) It seems that the second "$=$" on page 30 should be "$\leq$".

#Questions
1) A claim in this paper is that Accelerated-DP-SRGD beats the prior SoTA in terms of gradient complexity and batch gradient steps. In the non-private setting, can Nesterov's accelerated SGD achieve this effect, or is this improvement specific to the differential privacy setting?
2) In Section 1.1, you mention using DP matrix factorization to approximate the SRGs, but I couldn't find where this technique is used. Could you point it out?
3) What is the difference between the assumption that $\mathbf{x}^\dagger \in \mathcal{C}$ and the assumption of the realizable regime?
4) In general, when bounding the $\ell_2$-sensitivity, an algorithm without perturbation should be used. Why in the proof of Lemma 11, $\|\mathbf{b}_{t-1}\|_2$ is used when bounding $\\|\\mathbf{x}\_t-\\mathbf{x}\_{t-1}\\|$?
5) In the last equation of the first paragraph on page 24, why does $\sqrt{\frac{\sqrt{n}T}{c_{DP}B}}=\sqrt{c_{DP}}$ hold?
6) In Corollary 9, is the $polylog(n,1\/\\delta)$ term omitted? If so, I think $\widetilde{O}(\cdot)$ should be used.

**Paper Award:**

No

---

> ### Author Response · Authors · 2024-11-22
> **Response to reviewer Ggyu**
>
> Thanks to the reviewer for their suggestions on improving the paper. We will take these into account in a revision to improve the clarity and readability. Below we give answers to the reviewer's questions:
>
> **In the non-private setting, can Nesterov's accelerated SGD achieve this effect, or is this improvement specific to the differential privacy setting?**
> * Yes. Stochastic Nesterov’s acceleration does achieve better sample complexity than SGD. One way to see that is by setting b_max=0 (corresponds to the non-private setting) in Theorem 1. The non-stochastic term (or the optimization error) converges as 1/T^2 which is not feasible with standard SGD.
>
> **In Section 1.1, you mention using DP matrix factorization to approximate the SRGs, but I couldn't find where this technique is used. Could you point it out?**
> * This should say binary tree mechanism (which is a specific instantiation of DP-MF), we will correct this for consistency throughout the paper.
>
> **What is the difference between the assumption that $x^\dagger\in C$ and the assumption of the realizable regime?**
> * We believe these are the same; we will change the text to call this the realizable regime explicitly, to avoid confusion.
>
> **Why in the proof of Lemma 11, $||b_t||$ is used when bounding $||x_t - x_{t-1}||$?**
> * $x_t - x_{t-1}$ depends on $b_{t-1}$, so we cannot get an unconditional bound on $||x_t - x_{t-1}||$ better than $||C||$ without using a bound on $b_{t-1}$. The way we do this without violating privacy is to observe that with high probability all $b_t$ are bounded, and we can absorb the low-probability failure event into the DP $\delta$.
>
> **In the last equation of the first paragraph on page 24, why does $\sqrt{\frac{\sqrt{n}T}{c_{DP}B}} = \sqrt{c_{DP}}$ hold?**
> * This follows from the choice of parameters in (19). More explicitly, $T = c_{DP} n^{1/4}$ and $B = n^{3/4} / c_{DP}$, so we have $\sqrt{\frac{\sqrt{n}T}{c_{DP}B}} = \sqrt{\frac{\sqrt{n}(c_{DP}n^{1/4})}{c_{DP}(n^{3/4} / c_{DP})}} = \sqrt{c_{DP}}$. We will add this extra line to the proof to make it easier to follow.
>
> **In Corollary 9, is the polylog(n,1/δ) term omitted? If so, I think $\tilde{O}(\cdot)$ should be used.**
> * The bound in Corollary 9 is $c_{DP}^2$ instead of $c_{DP}$ (which $\sigma$ is proportional to in that corollary), and the extra $c_{DP}$ absorbs the polylog terms. We will make the calculation of the bound in the proof more explicit to make this clearer.

---

> > ### Comment · Reviewer_Ggyu · 2024-11-25
> >
> > Thank the authors for the detailed responses. My concerns have been addressed.

---

### Official Review · Reviewer_ke9a · 2024-11-15
**Improved parallel depth for private SCO under strong smoothness**

**Rating:** 6
**Confidence:** 4

**Review:**

The paper studies a "batch query" version of the classic DP-SCO problem, where the goal is to design a DP-SCO algorithm that 1) achieves the optimal learning error, 2) uses linear gradient query complexity, and 3) uses sublinear "query depth", i.e., the number of batches of gradient queries submitted. The main result of the paper is an algorithm using n^{1/4} batches in the realizable case (i.e., global optimizer in the constraint set), and otherwise n^{1/2} batches. To achieve this result the authors assume a fairly stringent smoothness bound of O(L/D) where L is the Lipschitz parameter and D is the diameter. Previously, it was known how to achieve goals 1 and 2 above assuming a smoothness bound of O(L * sqrt{n} / D) [FKT20]. Intuitively, O(L/D) is the "best case smoothness", as then the gradient norm can change by an additive L over the domain. Another side effect is that the algorithm doesn't use contractivity properties to argue about privacy as in [FKT20], instead directly controlling sensitivity, so it stays private for non-convex problems (albeit with unclear utility bounds).

The key idea is a tighter analysis of the noise sensitivity of AGD, by using smoothness to control gradient sizes, along with more careful gradient aggregation scheme (i.e., the binary tree mechanism of [DNPR10]). The authors also need to use the utility guarantees to bound the eventual sensitivity of their algorithm, which is a nice technical observation.

The paper is reasonably well-written, and studies a fairly well-motivated problem. Perhaps the biggest weakness is the fact that they have to essentially assume the strongest possible smoothness bound in their setting (stronger than prior works by a sqrt{n} factor). One thing that I was curious about is -- does the paper degrade gracefully at all if the smoothness is a bit worse than O(L/D), i.e., if it's instead LC/D, can you lose some poly(C) in either the query complexity or error but still get a parallel speedup?

Additionally, I feel like the paper does not highlight its technical connection to the existing literature on 1) parallel SCO, and 2) noise-tolerant acceleration, very clearly. For example, the "batch query complexity" in the paper is essentially the highly parallel model of computation introduced by Nemirovski in "On parallel complexity of nonsmooth convex optimization", which has a fairly extensive literature. Some relevant papers are discussed in the paper, but many such works, including e.g., "Randomized smoothing for stochastic optimization", "Complexity of highly parallel non-smooth convex optimization", and the most recent state-of-the-art "Closing the Computational-Query Depth Gap in Parallel Stochastic Convex Optimization" are not. The last paper has a survey of this literature that might be good to take a look at. In a similar vein, while I would believe that the exact statement of Theorem 3 has not appeared in the literature before, there are a fair number of papers that study stochastic or adversarial variants of Nesterov's acceleration. For example, "Optimal stochastic approximation algorithms for strongly convex stochastic composite optimization i: A generic algorithmic framework" contains a pretty similar analysis, and another example is "On Acceleration with Noise-Corrupted Gradients".

Overall, I think the paper makes an interesting enough conceptual contribution to be above the bar for acceptance, as I'm not even sure if the problem of parallelizing DP-SCO has really been studied before, and is a very reasonable direction of exploration for practical reasons.

Some more minor comments:
1. The discussions of "convexity for privacy" are somewhat confusing when first presented, because the problem is explicitly called DP-SCO. What I think the authors mean is that a bunch of previous linear query complexity algorithms all used that smooth, convex gradient steps are contractive, and their analysis does not. I feel like clarifying this early on would make it simpler to understand.
2. Could the authors discuss whether this zero-out DP notion is necessary / how to modify it for more standard definitions?
3. The FKT20 line in Table 1 is a bit confusing, as the gradient complexity is n but the steps are O~(n). Certainly it cannot be more than n.
4. bluntly <- crudely? (Page 5)
5. stochastcic <- stochastic (Page 6)
6. The comparison to [CJJLLST23] on Page 6 was very hard for me to understand. First of all, the authors state the rate of [CJJLLST23] as min(n, n^2/p) + min(np^2/eps, n^{4/3}eps^{1/3}), but the rate in [CJJLLST23] is actually min(...) + min((np)^{2/3}/eps, ...) The authors then say their overall gradient oracle complexity is O~(sqrt{n}) -- do you mean your batch complexity? Also, for sufficiently large n >> p^2, it seems the [CJJLLST23] algorithm also achieves linear gradient query complexity. Maybe the entire paragraph was meant to compare the batch complexity? (On Page 3, p <= eps^{3/2} is quoted as a transition for [CJJLST23], which was similarly confusing.)
7. Section 1.1 felt quite out of place (and largely redundant).
8. Discussion at top of Page 11, see the previous references on stochastic / noisy acceleration.
9. I think Lemma 8 has an incomplete sentence in its statement.

**Paper Award:**

No

---

> ### Author Response · Authors · 2024-11-22
> **Response to reviewer ke9a**
>
> Thanks to the reviewer for their feedback and questions. We address questions and comments below, and will address smaller comments such as typos in a revision.
>
> **One thing that I was curious about is -- does the paper degrade gracefully at all if the smoothness is a bit worse than O(L/D), i.e., if it's instead LC/D, can you lose some poly(C) in either the query complexity or error but still get a parallel speedup?**
> * We believe the result in the case $x^* \in \mathcal{C}$ can decay as poly(C) in both utility and query complexity, up to $C = n^{1/4}$. For the proofs of Lemma 6 and Lemma 7, we effectively require $C^2 \leq c_{DP} \leq n^{1/2}$ (both constraints come from the last inequality in the proof of Lemma 7; since this is used to prove a single induction step that is chained $T$ times, even constant slack in this inequality is not tolerable for our proof). Our batch gradient complexity is linear in $c_{DP}$ (eq. (19)) and our final error bound is linear in $||b_t||$ which is linear in $c_{DP}^2$. Putting it all together, we would pay $C^2$ in the runtime and $C^4$ in the error bound.
> * For our result not assuming $x^* \notin \mathcal{C}$, it is more straightforward - Lemma 11 and Corollary 12 both hold for general $C$, and $\beta$ and $Q$ in the utility bound of Theorem 3 increase linearly in $C$, so our final error bound increases by $C$.
>
> **Additionally, I feel like the paper does not highlight its technical connection to the existing literature on 1) parallel SCO, and 2) noise-tolerant acceleration, very clearly.**
> * Thank you for the guidance with the prior work. We will definitely map the definition of the computation model to that in Nemirovski’s paper. While we compared to Carmon et al. (which is an immediate prior work to Jambulpati et al.), we failed to discuss the connections mentioned in Jambulpati et al., and the references therein. We will update the draft to discuss these connections.
>
> **While I would believe that the exact statement of Theorem 3 has not appeared in the literature before, there are a fair number of papers that study stochastic or adversarial variants of Nesterov's acceleration...**
> * It is worth mentioning that one crucial aspect that our algorithm and the analysis differ from the prior work is that to obtain a strictly single pass/epoch algorithm, we not only had to analyze an adversarial variant of Nesterov’s method, we also had to combine it Stochastic-Recursive Gradients. In particular, in Theorem 3 we needed the variance in the gradient differences to be bounded, rather than in the stochastic gradients itself. We could not find a direct reference for Theorem 3 that has the right parameterization in terms of the $\eta_t$’s (which can later be tuned to), hence we proved it from scratch. In future versions of the paper, we will surely acknowledge prior works for looking at adversarial versions of Nesterov’s acceleration.
>
> **The discussions of "convexity for privacy" are somewhat confusing when first presented...**
> * The reviewer is correct, our main improvement is that contractivity is not used in our privacy argument which means it can be applied to arbitrary (i.e., non-convex) losses via clipping. We will definitely add a note describing where in the prior work convexity was important while stating they required convexity for privacy.
>
> **The FKT20 line in Table 1 is a bit confusing, as the gradient complexity is n but the steps are O~(n). Certainly it cannot be more than n.**
> * Sorry for the confusion. The number of steps is actually smaller than n, but only by polylog n factors. Hence we used the $\tilde\Theta$ notation. We will instead write an explicit $\Theta(\cdot)$ bound for this result to avoid confusion.
>
> **The comparison to [CJJLLST23] on Page 6 was very hard for me to understand...**
> * Thank you for pointing out the typo in stating the bound of [CJJLLST23]. Their bound is rightly what the reviewer stated. You are also right, we meant to say that our batch gradient complexity (which [CJJLLST23] calls query depth) is better, i.e.,   $\tilde O(\sqrt{n})$ as opposed to $\tilde{O}(n)$. Also the bound on page 3 should have been $p <= n^{1/2}\epsilon^{3/2}$. (There is a $\sqrt n$ term missing.) Hope this clarifies the confusion.
>
> **Section 1.1 felt quite out of place (and largely redundant).**
> * Section 1.1 was just to summarize the contribution. We will tersen it and make it clear it is just meant as a summary.
>
> **Discussion at top of Page 11, see the previous references on stochastic / noisy acceleration.**
> * We will address it as mentioned in the previous response.
>
> **I think Lemma 8 has an incomplete sentence in its statement.**
> * We will fix the typo. It now says “including $(\mu \sqrt{2 \ln(1.25/\delta)}, \delta)$-DP for all $\delta > 0$”.

---

### Meta-Review · Area_Chair_Eog3 · 2024-12-13

**Recommendation:** Accept
**Confidence:** 4

**Metareview:**

This paper studies a practically-motivated aspect of DP-SGD, which has not been addressed nontrivially before in previous literature: its adaptive complexity (i.e., parallel query depth). All reviewers agree that the paper makes interesting technical and conceptual contributions, and are in favor of acceptance. The reviewers also raised a number of weaknesses in the clarity of writing and contextualizing the literature, which were adequately addressed in the rebuttal period. I am recommending the acceptance of this paper at ALT.

**Paper Award:**

No